# Seasonal streamflow forecasting in the Upper Indus Basin of Pakistan: an assessment of methods

Stephen P. Charles[1], Quan J. Wang[2], Mobin-ud-Din Ahmad[3], Danial Hashmi[4], Andrew Schepen[5], Geoff Podger[3] and David E. Robertson[6]

[1]CSIRO Land and Water, Floreat, 6014, Australia
[2]The University of Melbourne, Parkville, 3010, Australia
[3]CSIRO Land and Water, Canberra, 2601, Australia
[4]Water and Power Development Authority, Lahore, Pakistan
[5]CSIRO Land and Water, Dutton Park, 4102, Australia
[6]CSIRO Land and Water, Clayton, 3168, Australia

*Correspondence to*: Stephen P. Charles (Steve.Charles@csiro.au)

**Abstract.** Timely and skilful seasonal streamflow forecasts are used by water managers in many regions of the world for seasonal water allocation outlooks for irrigators, reservoir operations, environmental flow management, water markets and drought response strategies. In Australia, the Bayesian joint probability (BJP) statistical approach has been deployed by the Australian Bureau of Meteorology to provide seasonal streamflow forecasts across the country since 2010. Here we assess the BJP approach, using antecedent conditions and climate indices as predictors, to produce Kharif season (April-September) streamflow forecasts for inflow to Pakistan's two largest Upper Indus Basin (UIB) water supply dams, Tarbela (on the Indus) and Mangla (on the Jhelum). For Mangla, we compare these BJP forecasts to (i) ensemble streamflow predictions (ESP) from the snowmelt runoff model (SRM) and (ii) a hybrid approach using the BJP with SRM-ESP forecast means as an additional predictor. For Tarbela, we only assess BJP forecasts using antecedent and climate predictors as we did not have access to SRM for this location. Cross validation of the streamflow forecasts show that the BJP approach using two predictors (March flow and an ENSO climate index) provides skilful probabilistic forecasts that are reliable in uncertainty spread for both Mangla and Tarbela. For Mangla, the SRM approach leads to forecasts that exhibit some bias and are unreliable in uncertainty spread, and the hybrid approach does not result in better forecast skill. Skill levels for Kharif (April-September), early Kharif (April-June) and late Kharif (July-September) BJP forecasts vary between the two locations. Forecasts for Mangla show high skill for early Kharif and moderate skill for all Kharif and late Kharif, whereas forecasts for Tarbela also show moderate skill for all Kharif and late Kharif, but low skill for early Kharif. The BJP approach is simple to apply, with small input data requirements and automated calibration and forecast generation. It offers a tool for rapid deployment at many locations across the UIB to provide probabilistic seasonal streamflow forecasts that can inform Pakistan's basin water management.

## 1 Introduction

The Asian Development Bank rates water security in Pakistan as 'hazardous' (the lowest of five classes), ranking it 46[th] out of 48 countries in the Asia-Pacific region, with only Kiribati and Afghanistan ranked lower (Asian Development Bank, 2016). Other studies confirm Pakistan's relatively high levels of exploitation of river flows and groundwater, associated water stress and resultant exposure to climate change (Döll et al., 2009;Wada et al., 2011;Schlosser et al., 2014;Kirby et al., 2017). Given the high demands on the main water source, the Indus River, its year to year flow variability has a significant impact on security of supply in the Indus Basin Irrigation System (IBIS) of Pakistan. Better management outcomes could be achieved if a reliable understanding of Kharif (summer, April-September) water availability at the beginning of the season were available. This would improve IBIS water allocation planning, a critical need given the highly seasonal flows (~80% annual flow occurs in the Kharif season), limited storage capacity (10% of inflows) and increasing water demand for agriculture and energy production. Thus we assess methods for providing seasonal streamflow forecasts for the two largest water supply dams, Tarbela (on the Indus) and Mangla (on the Jhelum), in the Upper Indus Basin (UIB) of Pakistan.

Seasonal streamflow forecasts can be a valuable source of information for water resource managers (Chiew et al., 2003;Anghileri et al., 2016), with both statistical and dynamical forecasting approaches developed and implemented internationally (Yuan et al., 2015). Sources of seasonal streamflow predictability come from initial hydrological or antecedent conditions (e.g. water held in storage in a catchment, in the soil, as ground water, in surface stores, or as snow/ice) and also from the skill of seasonal climate forecasts (Bennett et al., 2016;Doblas-Reyes et al., 2013;Li et al., 2009;Shukla and Lettenmaier, 2011;Shukla et al., 2013;van Dijk et al., 2013;Koster et al., 2010;Wood et al., 2015;Yossef et al., 2013). Statistical approaches relate antecedent catchment conditions and/or climate indices to streamflow using techniques such as multiple linear regression (Maurer and Lettenmaier, 2003). Statistical approaches require predictor-predictand records of sufficient length to determine robust relationships, stationarity in the relationships, and rigorous cross-validation to avoid over-fitting or an inflated skill assessment (Robertson and Wang, 2012;Schepen et al., 2012). Dynamical approaches use hydrological models initialised with observed inputs up to the beginning of the forecast season (to account for antecedent conditions) that can be driven either by historical or modelled climate inputs (Yuan et al., 2015;Zheng et al., 2013). For example, in Ensemble Streamflow Prediction (ESP), hydrological models are driven by each historical season's precipitation and temperature series to produce an ensemble of flow forecasts, with this ensemble providing a distribution of plausible flows for the forecast period (Wood and Lettenmaier, 2008). ESP forecasts can also be used as an input predictor to statistical techniques (Robertson et al., 2013). Dynamical (i.e. climate model) forecasts of precipitation and temperature are often not sufficiently skilful in this region. For example, Kim et al. (2012) assessed retrospective seasonal forecasts of the Asian summer monsoon from ECMWF System 4 (Molteni et al., 2011) and NCEP CFSv2 (Saha et al., 2014), finding low skill for precipitation prediction and poor simulations of the Indian summer monsoon circulation. Cash et al. (2017) assessed monthly North American Multi-Model Ensemble (Kirtman et al., 2013) hindcasts initialised May 1 for May to November for South Asia, including the mountainous areas of Afghanistan and Pakistan. They concluded that the multi-model ensemble mean temperature and precipitation forecasts, while

generally exceeding the skill of any individual model, provided little benefit over climatology. Given these findings, we have not investigated the use of dynamical forecasts in this assessment. Alternatively, statistically-based forecast methods using robust relationships between climate drivers, antecedent catchment conditions and resultant streamflow can be valuable research and management tools when properly implemented (Plummer et al., 2009;Schepen et al., 2016). Thus in this assessment, we assess three statistically-based forecast options for their practical feasibility in developing seasonal streamflow forecasting models for the study region:

1. A statistical approach using the Bayesian joint probability (BJP) model with predictors accounting for antecedent basin conditions and climate drivers;
2. An ESP approach using the snowmelt runoff model (SRM); and,
3. A hybrid approach using option (1) with an additional predictor – the mean ESP forecasts from (2).

The study is reported as follows: Section 2 outlines the study area, details of the case studies and data used, and climate influences; Section 3 presents the BJP statistical approach, the SRM-ESP approach, and the verification metrics used to assess forecast skill, bias, reliability and robustness; Section 4 presents the results of the BJP and SRM skill scores and performance diagnostics. Section 5 discusses the performance of the forecast approaches, and Section 6 concludes with the main findings and recommendations.

## 2 Case Study and Data

### 2.1 Upper Indus Basin

Pakistan's water supply, crucial for its extensive irrigated agriculture industry, hydropower generation, and industrial and municipal water supply, is predominantly sourced from Indus river flow, with groundwater a secondary although important contributor to most demands (with the exception of hydropower). The glaciated and snow covered sub-basins of the UIB, encompassing glaciated headwater catchments within the northern Hindu-Kush, Karakoram and western Himalayan mountain ranges, dominate water generation within the Indus Basin (Alford et al., 2014). The UIB's tributaries include the Indus at Kharmong: Shigar, Shyok and Astore in the Karakoram Himalaya, the Jhelum, Chenab, Ravi and Sutlej in the western Himalaya, and the Hunza, Gilgit, Kabul, Swat and Chitral in the Hindu Kush mountains (Figure 1). These basins can be classified as having a flow regime that is either glacier-melt dominated (Hunza, Shigar and Shyok) or snow-melt dominated (Jhelum, Kabul, Gilgit, Astore and Swat) (Hasson et al., 2014). The predominant source of flow in the UIB is snowmelt, with glacier melt a secondary source, with 80% of flow occurring during the June-September summer period. Interannual flow variability is thus controlled by two processes, snow accumulation as determined by winter precipitation and temperature and meltwater generation as determined by summer temperatures. Hence snowmelt-generated flow is a function of winter precipitation and temperature and also summer temperature, whereas glacier melt is primarily a function of summer temperature, although glacier melt is also influenced by snow cover (Charles, 2016).

Inflows to two major reservoirs, Tarbela Dam on the Upper Indus and Mangla Dam on the Jhelum River, a major tributary to the Indus system, are investigated (Figure 1). Daily inflow data from 1975 to 2015 was obtained from Pakistan Water and Power Development Authority (WAPDA). Figure 2 presents the seasonal hydroclimatic cycle for these two basins, showing double-peaked (winter and summer) precipitation with inflow peaking in May for Mangla and July for Tarbela. The Tarbela Dam on the main stem of the Indus is one of the largest individual storage in the UIB, crucial for hydropower generation and irrigation supply (Ahmed and Sanchez, 2011). Annual inflows to Tarbela constitute 70% melt water, of which snowmelt contributes 44% and glacial melts contribute 26% (Mukhopadhyay and Khan, 2015). The Mangla Dam on the Jhelum River is (since enlargement) a similar size storage to Tarbela and one of the most important resources in Pakistan for electricity generation and water supply for irrigation (Mahmood et al., 2015). For the Jhelum, the area upstream of Mangla is reported as 33,500 km$^2$ with an elevation range from 300 m to 6285 m and mean of nearly 2,400 m, the relatively low altitude ensures that there is only 0.7 % coverage by glaciers or perennial snow according to GLIMS glacier database as cited by Bogacki and Ismail (2016). In contrast, the Indus upstream of Tarbela is over five times larger (173,345 km$^2$) with higher elevation (to 8,238 m, as reported by Immerzeel et al. (2009)) and 11.5% covered by perennial glaciers (Ismail and Bogacki, 2018), such that (as noted above and evident in Figure 2, showing inflow exceeding precipitation) glacier ice melt is a significant contribution to annual flow.

## 2.2 Climate influences

Useful climate indices should relate to the weather prior to the forecast season, providing an indication of snow accumulation, and also to the weather within the forecast season, influencing temperature and hence snow and glacier melt rates. A literature review identified indices related to the North Atlantic Oscillation (NAO) and El Niño Southern Oscillation (ENSO) as the most likely to provide skill for the UIB (Charles, 2016). These both influence the direction of prevailing winds bringing moisture into the region and thus determine precipitation and temperature conditions influencing the depth and areal extent of snow accumulation in the winter and early spring preceding the Kharif (April-September) high-flow season.

The NAO is a measure of the strength of the pressure gradient between the subtropics and polar regions in the north Atlantic, representing a dominant source of variability in circulation and winds influencing the region (Hurrell, 1995;Bierkens and van Beek, 2009). It has a direct influence on the interannual variability of the westerly winds (westerly disturbances) and their water content traversing Europe, the Mediterranean and the Middle East region into the mountains of the UIB (Yadav et al., 2009a;Syed et al., 2010;Filippi et al., 2014). Indices of the NAO have been related to: UIB station winter precipitation (Archer and Fowler, 2004;Afzal et al., 2013;Filippi et al., 2014); western Indus basin's winter snow cover and station precipitation (Hasson et al., 2014); Pakistan station temperature (del Río et al., 2013); and winter precipitation in northwest India (Kar and Rana, 2014).

ENSO is a dominant pattern of multi-year variability driven by ocean-atmosphere interactions in the tropical Pacific (Wolter and Timlin, 2011), influencing climate globally including the variability of both western disturbances and monsoon processes experienced by the region. The commonly used SOI (Southern Oscillation Index) has been related to: winter Hindu Kush

Himalayan region precipitation (Afzal et al., 2013); Indian Summer Monsoon Precipitation (Ashok et al., 2004;Ashok and Saji, 2007); central southwest Asian winter precipitation (Syed et al., 2006); Pakistan station temperature (del Río et al., 2013); and northwest India winter precipitation (Kar and Rana, 2014). Stronger links have been reported between ENSO, western disturbances and interannual winter precipitation variability for recent decades (Yadav et al., 2009a;Yadav et al., 2009b).

## 3 Methods

### 3.1 BJP forecasting models

The statistical seasonal forecasting model used is the Bayesian joint probability (BJP) approach of Wang et al. (2009). The BJP offers state-of-the-art capabilities for developing seasonal forecast models that optimally utilise information available on antecedent catchment conditions, large-scale climate forcing (through climate indices) and flow forecast scenarios from hydrological models (Robertson et al., 2013;Robertson and Wang, 2012;Schepen et al., 2012;Wang and Robertson, 2011). The BJP models simulate predictor-predictand relationships using conditional multivariate normal distributions, with predictor and predictand data transformed to normal using either a log-sinh (Wang et al., 2012b) or Yeo-Johnson (Yeo and Johnson, 2000) transformation. BJP parameters are inferred using Markov Chain Monte Carlo methods (MCMC) to account for parameter uncertainty, which can be due to factors such as short data records. Probabilistic (ensemble) forecasts are produced by generating samples from the estimated conditional multivariate normal distributions. When predictor-predictand relationships are weak, the BJP produces reliable forecasts that approximate climatology. The full technical details of the BJP modelling approach are presented in Wang et al. (2009) and Wang and Robertson (2011).

### 3.2 SRM forecasting model

The Snowmelt Runoff Model (SRM) of Martinec et al. (2008) has been used in several studies in the basin (Butt and Bilal, 2011;Romshoo et al., 2015;Tahir et al., 2011;Bogacki and Ismail, 2016;Ismail and Bogacki, 2017, 2018). WAPDA has procured a version of SRM implemented in MS-Excel©, 'ExcelSRM', and for their 2012 case study for Jhelum inflow into the Mangla Reservoir (NESPAK et al., 2012), ExcelSRM was calibrated using data for 2003 to 2010 and subsequently validated against inflows for 2000 to 2002 and 2007 and 2011(Bogacki and Ismail, 2016). In contrast to the probabilistic forecasts produced by the BJP, the SRM is a deterministic model and so produces a single forecast for a given set of inputs.

Given the inadequacy of seasonal meteorological forecasts for the region (Bogacki and Ismail, 2016;Cash et al., 2017;Ismail and Bogacki, 2018), an ESP approach is used to forecast a range of possible Kharif season inflows. That is, SRM is initiated with end of March observed snow cover, and then run to produce six-month Kharif-season scenarios using the P and T inputs from each year in the available historical record, together with the Modified Depletion Curve approach (Rango and Martinec, 1982) to simulate snow cover depletion as a function of that scenario-year's degree-days series. This approach results in an ensemble of simulated inflows and, as well as assessing the SRM-ESP forecasts themselves (from a research version of

ExcelSRM we obtained in 2015; i.e. not the current version used operationally), this study has used the mean of the ensemble members for each year as an additional predictor series for input to the BJP (option 3 as outlined in the Introduction).

## 3.3 Verification

BJP forecast model performance is verified using leave-one-out cross-validated results (Wang and Robertson, 2011). That is, to avoid artificially inflating the skill, for each Kharif season the calibration and assessment does not use that season's data for BJP parameter estimation. The cross-validated BJP forecast performance was assessed for 1975-2015 (41 seasons), with the BJP models calibrated on a seasonal basis (i.e. 40 data points) using 1000 MCMC samples for each of the leave-one-out calibrations.

As noted, the BJP can also use hydrological model simulated flow as an additional predictor (Robertson et al., 2013) and in this case we have SRM simulations for the Mangla inflow for a subset of the investigation period (2001-2015). The SRM is an exception to the leave-one-out cross validation as it is calibrated using all data for the 2003 to 2010 period, with parameters manually tuned "… *in order to keep parameters at smooth values and to maintain a reasonable trend in time*." (NESPAK et al., 2012). The mean flow simulation obtained from driving each year's SRM with all year's available precipitation and temperature are therefore not independent forecasts and so it is not surprising that for 2003 to 2010 period the SRM forecasts can be closer to the observed flows than the median cross-validated BJP estimates. When used as a predictor to the BJP, SRM forecasts are applied with leave-one-out cross validation i.e. for each year in the 2001-2015 period all of the other 14 year's simulations are used to produce an ESP with the resulting mean used as a predictor for that year. Note the BJP is able to extract skill from biased dynamic hydrological model forecasts, as long as the hydrological model simulation bias is systematic and stationary (i.e. not random or with a trend).

Verification assesses the overall skill and the bias, reliability and robustness of the forecasts. This includes assessing whether the bias and reliability of the forecasts varies for different periods of the record (temporal stability) or for different event sizes, e.g. whether there is a limitation in forecasting high- or low-flow seasons. Skill scores, quantifying the skill of the forecasts, allow the direct comparison of the performance of forecasting models that use different sets of predictors. Two common skill scores used here are the root mean squared error (RMSE) that assesses the forecast median and the continuous ranked probability score (CRPS) that assesses the reduction in error of the whole forecast probability distribution (Robertson and Wang, 2013). The skill scores are reported as percentage reductions in error scores of the forecasts relative to the observed historical (climatological) median, for RMSE, and relative to the full distribution of the observed historical (climatological) events, for CRPS. The 'sharpness' of a probabilistic forecast distribution (i.e. a narrower peaked distribution rather than a wide, flat distribution) is also a characteristic relevant to forecast skill (Gneiting et al., 2007). Sharp forecasts with narrow forecast intervals reduce the range of possible outcomes that are anticipated, increasing their usefulness for decision makers (Li et al., 2016). This skill can be quantified, for example, as the percentage reduction in the inter-quartile range (IQR) between

the forecast's distribution and the observed historical (climatological) distribution (Crochemore et al., 2017). RMSE. CRPS and IQR skill scores are interpreted as[1]:

- 0 is considered to be a forecast with no skill (equivalent skill to predicting using historical averages or historical reference);
- less than 5 is considered to be a forecast with very low skill;
- 5-15 is considered to be a forecast with low skill;
- 15-30 is considered to be a forecast with moderate skill; and,
- greater than 30 is considered to be a forecast with high skill.

Reliability refers to the statistical similarity between the forecast probabilities and the relative frequencies of events in the observations, which can be verified using probability integral transforms (PITs). The PIT represents the non-exceedance probability of observed streamflow obtained from the CDF of the ensemble forecast. If the forecast ensemble spread is appropriate and free of bias then observations will be contained within the forecast ensemble spread, with reliable forecasts having PIT values that follow a uniform distribution between 0 and 1 (Laio and Tamea, 2007). Thus PIT plots are an efficient diagnostic to visually evaluate whether the forecast probability distributions are too wide or too narrow or are biased (under or over estimating) in their prediction of the observed distribution (Wang and Robertson, 2011). As outlined by (Thyer et al., 2009), PIT plot points falling on the 1:1 line indicate that the predicted distribution is a perfect match to the observed; observed PIT values of 0.0 or 1.0 indicate the corresponding observed data falls outside the predicted range, hence the predictive uncertainty is significantly underestimated; PIT values clustered around the midrange (i.e. a low slope in the 0.4 -0.6 uniform variate range) indicate the predictive uncertainty is overestimated; PIT values clustered around the tails (i.e. a high slope in the 0.4 -0.6 uniform variate range) indicate the predictive uncertainty is underestimated; and if PIT values at the theoretical median are higher than those of the uniform variate the predictions have an underprediction bias, and vice versa if they are lower than the uniform variate then the predictions have an overprediction bias.

## 4 Results

### 4.1 Skill scores

BJP models were trialled with combinations of predictors accounting for antecedent flow (flow immediately preceding the forecast season, i.e. March flow for the Kharif forecast) and NAO- or ENSO-based climate indices identified from the literature, as introduced in section 2.2 (Charles, 2016). MODIS (Hall et al., 2010) snow-cover area and GLDAS-2.1 (Rodell et al., 2004) snow-water equivalent, additional measures of antecedent conditions, were also assessed as potential predictors. A significant limitation is the shorter record lengths for MODIS and GLDAS, as available data for both start in 2000. This is of particular concern for the BJP's leave-one-out cross-validation, as using short records to identify dynamical mechanisms is susceptible to spurious skill. Correlation analysis, cognisant of the short 2000-15 period, found that these snow products have

---

[1] '*How are the skill score categories defined?*' from  http://www.bom.gov.au/water/ssf/faq.shtml

a similar or lower correlation with Kharif flow ($Q_{Kharif}$) compared to March flow ($Q_{March}$), and are relatively highly correlated with $Q_{March}$. Thus the limitation of short record length, lack of higher correlation with flow than that of the $Q_{March}$ predictor, and relatively high cross-correlation with $Q_{March}$, leads us to conclude that they would not be expected to provide additional skill as a predictor for the BJP.

SRM-ESP scenario-mean forecasts were an additional predictor trialled for Jhelum at Mangla. Higher skill was generally obtained for predictor combinations using a flow predictor (March flow) together with either the Multivariate ENSO index (MEI; http://www.esrl.noaa.gov/psd/enso/mei/index.html) (Wolter and Timlin, 1998) or the Southern Oscillation signal index (SSI; http://www.cgd.ucar.edu/cas/catalog/climind/soiAnnual.html) (Trenberth, 1984) as a climate predictor. The seasons of the trialled climate predictors (i.e. their time-lag preceding the forecast season) were selected based on their highest linear

correlations with flow (not shown). Table 1 presents cross-validated BJP forecast skill scores using the trialled combinations of antecedent flow and climate predictors for the Kharif season for Jhelum at Mangla, together with bootstrapped 10[th] to 90[th] percentile ranges to assess model uncertainty. These ranges were obtained by resampling 1000 random sequences of years of the same length as the observed record, i.e. with replacement, and calculating skill scores for each sample. Combinations including the SRM forecasts (ESP mean) as a predictor are included, however because the SRM results are only available for

the 15 year period 2001-2015, they are only providing skill during the 2001-2015 period when included as a BJP predictor for the full 41 year period (1975-2015). These results show:

- The antecedent predictor (March flow, $Q_{March}$) provides greater skill than any of the individual climate predictors used.
- Two-predictor models using $Q_{March}$ and the $SRM_{Kharif}$ predictor give poorer skill scores compared to using $Q_{March}$
alone.
- Two-predictor models using $Q_{March}$ and one climate predictor slightly improve (in most cases) the skill scores compared to using $Q_{March}$ alone.
- Given there is large uncertainty in skill scores we do not aim to select a 'best' model. However, as there are many models with positive skill (i.e. better than climatology) then using skilful models is plausible. Ideas on how to do
this are discussed in section 5.

Addition of the $SRM_{Kharif}$ predictor to the two-predictor models using $Q_{March}$ and one climate predictor does not improve skill scores. Table 2 presents the skill scores for the Kharif season forecasts for the Indus at Tarbela, also using the antecedent flow and climate predictors but without SRM forecasts in this case (as SRM was not available for Indus at Tarbela for this study). Similarly to the results for the Jhelum at Mangla, for the Indus at Tarbela BJP forecasts:

- The antecedent predictor (March flow, $Q_{March}$) provides greater skill than any of the climate predictors used.
- On the whole, a single climate predictor produces low skill compared to that obtained using $Q_{March}$, with a notable exception that the $MEI_{MayJun}$ (i.e. the year before) predictor produces skill scores comparable to those obtained using $Q_{March}$. The selection of $MEI_{MayJun}$ as a predictor is discussed further in Section 5.
- Two-predictor models using $Q_{March}$ and one climate predictor improve the skill scores compared to using $Q_{March}$
alone.
- Given there is large uncertainty in skill scores we do not aim to select a 'best' model. However, as there are many models with positive skill (i.e. better than climatology) then using skilful models is plausible. Ideas on how to do this are discussed in section 5.

In addition to calibration for the full Kharif season, BJP calibrations were also undertaken for the early Kharif (April-June) and the late Kharif (July-September) using the relevant flow and ENSO-based predictors (e.g. for late Kharif the June flow was used as an antecedent predictor). A comparison of the resulting skill scores are shown in Figure 3 for the BJP models that gave the highest skill gain relative to climatology for the Kharif, early Kharif and late Kharif periods. It is interesting to contrast the performance for the two locations, with Kharif and late Kharif giving similar results across the two locations whereas for early Kharif a marked difference is seen, with high skill for Jhelum at Mangla contrasting the low skill for Indus at Tarbela. The physical reasons for this contrast would require further investigation, with possible causes discussed in Section 5.

## 4.2 Performance diagnostics

Here we assess the cross-validated performance of forecasts from BJP models using an antecedent and climate predictor combination (option 1) selected on the basis of skill scores and, for Mangla, compare with the SRM-ESP forecasts (option 2). We do not compare results for the BJP models using the SRM-ESP mean as an additional predictor (option 3), as the addition of this predictor added little or no skill to BJP forecasts (Table 1).

We use PIT plots for verification of the reliability and robustness of the forecast probability distributions, to assess whether there are biases in the forecasts, or whether the forecast probability distributions are too wide or too narrow (Laio and Tamea, 2007). For reliable forecasts, the PIT values should follow a uniform distribution and hence follow a 1:1 line when plotted against a standard uniform variate. For the BJP forecasts for the full 1975-2015 period, both Jhelum at Mangla (Figure 4a) and Indus at Tarbela (Figure 5a) show reliability (i.e. forecast probability distributions are unbiased and of appropriate spread), evidenced by the forecast's PIT values plotting close to the 1:1 lines and within the Kolmogorov 5% significance band. Comparison of Jhelum at Mangla BJP and SRM-ESP forecasts, for the shorter 2001-2015 period for which SRM results are available, show a contrast between reliable BJP forecasts (Figure 6a) and biased SRM forecasts, including five values of 0 or 1 indicating that the SRM forecast distribution is too narrow (Figure 7a).

A feature of robust forecasts is their stability across the full period of record and range of flow magnitudes. Figure 4b and Figure 5b show a uniform spread of PIT values and hence stability across the full period of record, for Jhelum at Mangla and Indus at Tarbela BJP forecasts, respectively. Similarly, forecast stability across the range of flow magnitudes is verified by the uniformity of PIT values against forecast median, as shown in Figure 4c and Figure 5c for Jhelum at Mangla and Indus at Tarbela, respectively. For the Jhelum at Mangla BJP and SRM-ESP comparison, stability across time and flow magnitude are harder to assess given the short 15 year sample size. Figure 6b and c show reasonable stability of the BJP forecasts, although there is a trend over time for this part of the record. The equivalent SRM plots (Figure 7b and c) are not as robust, with (as noted previously) five values at 0 or 1 (from the chronological plot: 2001 is at 0, and 2010, 2012, 2014 and 2015 are at 1).

Robustness is also assessed by plotting forecast quantile ranges and observed flows against the forecast median (Figure 4d and Figure 5d) and chronologically (Figure 4e and Figure 5e, for Jhelum at Mangla and Indus at Tarbela respectively). These show that BJP forecasts reasonably account for the range of observed variability for both locations. The relatively less robust SRM-

ESP forecasts are shown in Figure 7d and e, again highlighting the overly narrow forecast distribution range for this version of SRM.

Overall, the performance shown in these figures highlight the reliability and robustness of the BJP forecasts for the Kharif season for Jhelum at Mangla and Indus at Tarbela. For the Jhelum at Mangla BJP and SRM-ESP forecasts for the shorter 2001-2015 period, the results contrast the reliable and robust BJP against some limitations from the SRM, which will be discussed further in the next section.

## 5 Discussion

The SRM forecasts are examples of the commonly applied ESP approach (Shi et al., 2008;Shukla and Lettenmaier, 2011;Wood et al., 2005). As such, Wood and Schaake (2008) note *"One strength of the ESP approach is that it accounts for uncertainty in future climate, which in some seasons is the major component of forecast uncertainty, by assuming that historical climate variability is a good estimate of current climate uncertainty. A weakness of the approach, however, is that when the uncertainty of the current ("initial") hydrologic state is a significant component of the overall forecast uncertainty …, the deterministic estimate of the forecast ensemble's initial hydrologic state leads to an overconfident forecast—that is, one having a spread that is narrower than the total forecast uncertainties warrant."*

This can be seen in the poor verification performance of SRM-ESP shown in Section 4.2, with the SRM-ESP from the 15 year sample unable to account for the observed range of flows, i.e. the ESP range is too narrow, even though in terms of overall RMSE and MAPE statistics the SRM-ESP-mean and BJP-median are comparable (18.7% RMSE and 14.3% MAPE for BJP-median; 18.4% RMSE and 12.7% MAPE for SRM-ESP-mean). Given that the observed climate of each individual year would, in most cases, be within the range of the ensemble of climate inputs used to produce the ESP, this indicates SRM formulation could be too strongly reliant on antecedent conditions at the beginning of the forecast season. An additional source of bias, as evidenced by the years of poorest performance being outside the years used for parameter estimation, could be over fitting with the model parameters tied too closely to the range of observed predictor-predictand relationships in the 2003-2010 calibration period.

The poorer BJP performance for Indus at Tarbela, as seen in the skill scores relative to Jhelum at Mangla (Figure 3), could be related to the differences in flow generation mechanisms. As the predictors are the source of skill in the statistical BJP approach, examination of correlations between the predictors and flow for the individual months within the Kharif season is insightful. Table 3 shows the (intuitively) expected pattern for Jhelum at Mangla of $Q_{March}$ having a maximum correlation with April flow (0.84) and then maintaining a relatively high correlation until July (0.62) before dropping off for August (0.12). A different process appears to be influencing Indus at Tarbela, as the initial highest $Q_{March}$ correlation with April flow (0.66) drops immediately to 0.18 for May before oscillating between 0.25 (August) to 0.55 (September) for subsequent Kharif months. Similarly, the climate predictor's correlations with the individual month's flows show more of a gradual reduction for Jhelum at Mangla (high for the first four months), whereas for Indus at Tarbela again an oscillatory relationship is seen. These

higher correlations in the late Kharif (relative to the early Kharif) for Indus at Tarbela would relate to the correspondingly higher relative skill scores shown in Figure 3 for late Kharif, corresponding to late-season glacier melt processes that are a significant component of the inflow to Tarbela but not Mangla (Mukhopadhyay and Khan, 2015). Future research could investigate whether dynamical seasonal forecasts of temperature have skill of relevance to forecasting glacier melt, however as noted above such skill has not been determined to date (e.g. Cash et al., 2017), and is beyond the scope of this assessment given our focus of developing practical and easily implementable forecast tools using readily available inputs.

It is also interesting to reflect on the relative performance of the NAO climate predictor, which does not provide any skill for inflow to Mangla (Table 1) but offers comparable skill to several of the ENSO indices trialled for Tarbela (Table 2). This indicates NAO may have some skill with regards to late season glacier melt. Overall, these results concur with investigations showing a stronger relationship between ENSO and precipitation and weaker relationship between NAO and precipitation in recent decades (Yadav et al., 2009a;Yadav et al., 2009b) resulting in the prevalence of ENSO as the better predictor of winter snowpack magnitude.

For Mangla, the predictor combination that gave the best Kharif season cross-validated skill scores included an ENSO-based predictor ($SSI_{March}$) immediately before the season (Table 1), which makes sense intuitively as it represents a climate driver of both the snow accumulation before and precipitation conditions during the Kharif season. In contrast, for Tarbela a much earlier ENSO-based predictor ($MEI_{MayJun}$, i.e. May-Jun the year before) provides higher skill scores than the equivalent predictor immediately before the season ($MEI_{FebMar}$) (Table 2). To try to understand the dynamical mechanism by which $MEI_{MayJun}$ is providing skill in forecasting $Q_{Kharif}$, we compared MEI correlations with GLDAS $SWE_{March}$, $Q_{March}$ and $Q_{Kharif}$. Results were inconclusive, and perhaps impeded by the short record lengths given SWE is only available from 2000, as while $MEI_{MayJun}$ has a higher correlation with $Q_{Kharif}$ than $MEI_{FebMar}$ (0.76 versus 0.63, respectively) it has a slightly lower correlation with $SWE_{March}$ (0.48 versus 0.52, respectively). Hence $MEI_{MayJun}$ does not appear to be a long-lead predictor of snow accumulation, and so the differences in skill scores may be due to spurious correlations. Therefore we recommend both this model and the $Q_{March}$ and $MEI_{FebMar}$ model be compared and assessed for future events. More generally, the skill score uncertainty ranges presented in Table 1 and Table 2 highlight that no 'best' forecast model can be selected for either basin. Attempting to select a best model would ignore model uncertainty and thus not make best use of forecast skill across the range of models trialled. To address this, probabilistic forecasts from multiple BJP models can be combined using Bayesian Model Averaging to produce combined forecasts with higher skill than that obtainable from any individual model (Wang et al., 2012a). Thus trialling a BMA approach is recommended, although it is beyond the scope of this current work.

**6 Conclusion**

This study has assessed the performance and practical feasibility of three options for producing Kharif (April-September) seasonal streamflow forecasts for the Jhelum River inflows to the Mangla Dam in the UIB of Pakistan: option 1, the BJP statistical forecasting technique; option 2, the SRM physically-based model run in ESP mode; and option 3, a hybrid of option

1 with the mean ESP forecasts from option 2 used as an additional predictor for input to the BJP. The option 1 BJP forecast model used antecedent catchment and climatic predictors, with the predictors selected based on BJP skill score performance. The selected predictors represent hydrological conditions immediately preceding the forecast season (i.e., flow of the preceding month – March in this case) and ENSO-based climate indices related to drivers of winter snow accumulation. For an additional comparison, the option 1 BJP approach was also undertaken for the Indus River inflows to the Tarbela Dam.

Overall findings were:

- The best performing BJP models for Tarbela and Mangla inflows are consistent in that both used an antecedent flow predictor and a climate predictor representing ENSO.

- For Tarbela the $Q_{March}$ and $MEI_{MayJun}$ model gave the best skill, however because we could not determine the dynamical mechanism(s) by which the relatively long lag between $MEI_{MayJun}$ influences snowpack accumulation and flow, we cannot rule out the possibility that the skill is due to spurious correlation. Therefore we recommend both this model and the $Q_{March}$ and $MEI_{FebMar}$ model be compared and assessed for future events and, more generally, that BMA be trialled in future research to combine the skill of multiple BJP models as, for example, undertaken in Australia in (Pokhrel et al., 2013).

- There are pragmatic benefits to selecting a BJP model using only antecedent and climate predictors, rather than including SRM mean ESP as an additional predictor even in cases when SRM does provide skill, given that flow and climate predictors are readily available and thus BJP forecasts are easily and quickly produced. The SRM, being a deterministic model, is a much more technically involved and data intensive approach to forecast generation (Bogacki and Ismail, 2016;Ismail and Bogacki, 2018).

- Cross-validated performance of the BJP seasonal forecasts for the 1975 to 2015 Kharif seasons, as shown in the diagnostic and verification statistics presented, highlight that the BJP produces forecasts that are statistically unbiased, robust and reliable. In contrast, the SRM-ESP forecasts show bias particularly for the most recent years outside the SRM calibration period, potentially indicating limitations with the SRM due to lack of cross-validated calibration and resultant over-fitting. Thus SRM-ESP forecasts are overly confident, underestimating the full uncertainty that is captured by the BJP approach.

- High skill was obtained for BJP forecasts of early Kharif flow for Jhelum at Mangla. Moderate skill was obtained for the full and late Kharif season forecasts for both Jhelum at Mangla and Indus at Tarbela. Lower skill was seen for early Kharif for Indus at Tarbela.

In future research, BJP forecast models could readily be developed and assessed for other tributaries, e.g. the Chenab and Kabul, subject to availability of flow data. This would allow an overall assessment of UIB flow forecasting for the major contributing basins. The present method used by the Indus River System Authority (IRSA) to forecast UIB Kharif streamflow is based on historical analogues. The IRSA use their database of the previous 60 years of flow to select years where the historic March flows are within 5% of the current March flow and use the corresponding historical Kharif flows (within 5%) for their forecast scenario. The selection of the historical scenario is also informed by forecasts from the Pakistan Meteorological

Department, forecasts provided by WAPDA (e.g. SRM forecasts), and present snow conditions in the catchment. The forecasts are continuously revised as the season progresses.

Sufficiently skilful BJP forecasts could also inform scenario selection, providing for the first time a probabilistic approach to forecasts in contrast to a single forecast as currently used. However probabilistic forecasts (such as a the BJP) can be

misinterpreted if they are unfamiliar to the water management professionals using them to inform decisions (Pagano et al., 2002;Ramos et al., 2013;Rayner et al., 2005;Whateley et al., 2015). Hence the successful transfer of BJP forecast tools to operational use within Pakistan would require guidance for building BJP models and generating forecasts, test cases with example results, face to face training, and on-going support.

*Competing interests.* The authors declare that they have no conflict of interest.

*Acknowledgements.* The Pakistan Mission of the Australian Department of Foreign Affairs and Trade provided funds for this study through the Sustainable Development Investment Portfolio Indus project. CSIRO provided additional support through its Land and Water portfolio. We thank the Pakistan Meteorological Department (PMD) for sharing datasets and the Pakistan

Water and Power Development Authority Glacier Monitoring Research Centre (WAPDA-GMRC) for sharing datasets and the SRM used in this collaborative work. Thanks also to Luis Neumann and Tony Zhao, CSIRO Land and Water, for help and advice. We express our gratitude to the two anonymous reviewers and the editor, Andy Wood, for their inputs which helped improve the paper. We also acknowledge helpful comments received from Muhammad Fraz Ismail regarding the SRM.

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

**Tables**

**Table 1: Cross-validated skill scores of BJP forecasts for Kharif season for Jhelum at Mangla. Bootstrap 10[th] to 90[th] percentile resampling ranges are shown in brackets.**

| Predictor combination | | | Skill scores 2001-2015 | | | | Skill scores 1975-2015 | | | |
|---|---|---|---|---|---|---|---|---|---|---|
| Flow[a] | Climate[b] | Model[c] | SSCRPS | | SSRMSE | | SSCRPS | | SSRMSE | |
| $Q_{March}$ | – | – | 21.0 | (-9.5 – 41.3) | 17.5 | (-13.8 – 39.6) | 26.9 | (15.7 – 35.8) | 26.0 | (14.5 – 35.3) |
| $Q_{March}$ | – | $SRM_{Kharif}$ | 15.8 | (-12.6 – 36.0) | 13.2 | (-13.2 – 32.4) | 25.6 | (15.2 – 34.9) | 25.4 | (15.0 – 34.2) |
| – | $NAO_{SepOctNov}$ | – | -3.2 | (-19.5 – 10.5) | -7.1 | (-22.6 – 7.3) | -0.6 | (-4.8 – 2.9) | -2.3 | (-7.3 – 1.8) |
| – | $MEI_{FebMar}$ | – | 14.7 | (-1.3 – 28.2) | 10.3 | (-5.8 – 26.2) | 14.2 | (5.8 – 21.8) | 11.7 | (3.1 – 19.9) |
| $Q_{March}$ | $MEI_{FebMar}$ | – | 22.6 | (-5.0 – 42.9) | 18.5 | (-9.0 – 40.7) | 25.5 | (13.8 – 35.0) | 24.7 | (13.0 – 33.9) |
| – | $MEI_{FebMar}$ | $SRM_{Kharif}$ | 24.1 | (5.2 – 38.2) | 21.2 | (1.6 – 36.2) | 17.7 | (7.6 – 26.5) | 15.4 | (4.5 – 24.4) |
| $Q_{March}$ | $MEI_{FebMar}$ | $SRM_{Kharif}$ | 18.0 | (-6.7 – 38.0) | 14.8 | (-10.0 – 34.8) | 23.9 | (12.5 – 34.0) | 23.2 | (11.5 – 32.6) |
| | $SSI_{March}$ | – | 11.4 | (-5.3 – 27.5) | 7.0 | (-9.2 – 27.9) | 7.9 | (0.6 – 14.8) | 6.0 | (-1.7 – 13.6) |
| $Q_{March}$ | $SSI_{March}$ | – | 24.3 | (-2.5 – 44.9) | 20.2 | (-7.2 – 42.5) | 26.2 | (15.7 – 35.8) | 25.1 | (14.4 – 34.5) |
| | $SSI_{March}$ | $SRM_{Kharif}$ | 24.2 | (2.2 – 41.3) | 20.5 | (-3.2 – 39.5) | 12.9 | (3.0 – 21.7) | 10.3 | (-0.4 – 19.6) |
| $Q_{March}$ | $SSI_{March}$ | $SRM_{Kharif}$ | 22.5 | (-5.2 – 42.5) | 19.1 | (-8.0 – 39.0) | 24.9 | (13.3 – 34.8) | 24.4 | (13.4 – 33.7) |
| | | | | | | | | | | |
| SRM Scenarios[d] | | | 25.3 | | 20.4 | | – | | – | |

[a] 1976-2015

[b] 1975-2015

[c] 2001-2015 SRM-ESP mean, with L1OCV

[d] 2001-2015 SRM-ESP with L1OCV

**Table 2: Cross-validated skill scores of BJP forecasts for Kharif season for Indus at Tarbela. Bootstrapped 10[th] to 90[th] percentile resampling ranges are shown in brackets.**

| Predictor combination | | | Skill scores 1975-2015 | | | |
|---|---|---|---|---|---|---|
| Flow[a] | Climate[b] | Model[c] | SSCRPS | | SSRMSE | |
| $Q_{March}$ | – | – | 16.6 | (9.0 – 24.0) | 18.9 | (11.8 – 26.0) |
| – | $NAO_{SepOctNov}$ [d] | – | 6.1 | (1.1 – 11.1) | 8.2 | (2.8 – 13.7) |
| $Q_{March}$ | $NAO_{SepOctNov}$ [d] | – | 18.8 | (11.9 – 27.0) | 21.0 | (14.2 – 28.4) |
| – | $MEI_{FebMar}$ | – | 7.6 | (2.5 – 12.7) | 10.3 | (4.2 – 15.9) |
| $Q_{March}$ | $MEI_{FebMar}$ | – | 18.6 | (9.9 – 25.9) | 20.8 | (12.0 – 28.1) |
| – | $MEI_{MayJun}$ [d] | – | 15.6 | (7.2 – 23.2) | 17.1 | (7.9 – 25.8) |
| $Q_{March}$ | $MEI_{MayJun}$ [d] | – | 25.0 | (15.1 – 33.6) | 25.0 | (14.2 – 34.6) |
| – | $SSI_{March}$ | – | 1.4 | (-1.6 – 4.6) | 4.9 | (0.3 – 9.0) |
| $Q_{March}$ | $SSI_{March}$ | – | 16.9 | (9.1 – 24.6) | 19.3 | (11.3 – 26.6) |

[a] 1976-2015

[b] 1975-2015

[c] No SRM for Indus

[d] noting lag to calendar-year before flow season

**Table 3: Correlation between the flow of the individual months of the Kharif season and the predictors used by BJP (those in bold significant at p < 0.05)**

| $Q_{Month}$ | Jhelum at Mangla | | Indus at Tarbela | |
|---|---|---|---|---|
| | $Q_{March}$ | $SSI_{March}$ | $Q_{March}$ | $MEI_{MayJun}$[a] |
| Apr | **0.84** | **-0.50** | **0.66** | **0.41** |
| May | **0.77** | **-0.41** | 0.18 | 0.22 |
| Jun | **0.62** | **-0.32** | **0.44** | 0.28 |
| Jul | **0.62** | **-0.38** | **0.34** | **0.42** |
| Aug | 0.12 | -0.03 | 0.25 | **0.37** |
| Sep | 0.18 | -0.27 | **0.55** | 0.30 |

[a]Year before

**Figures**

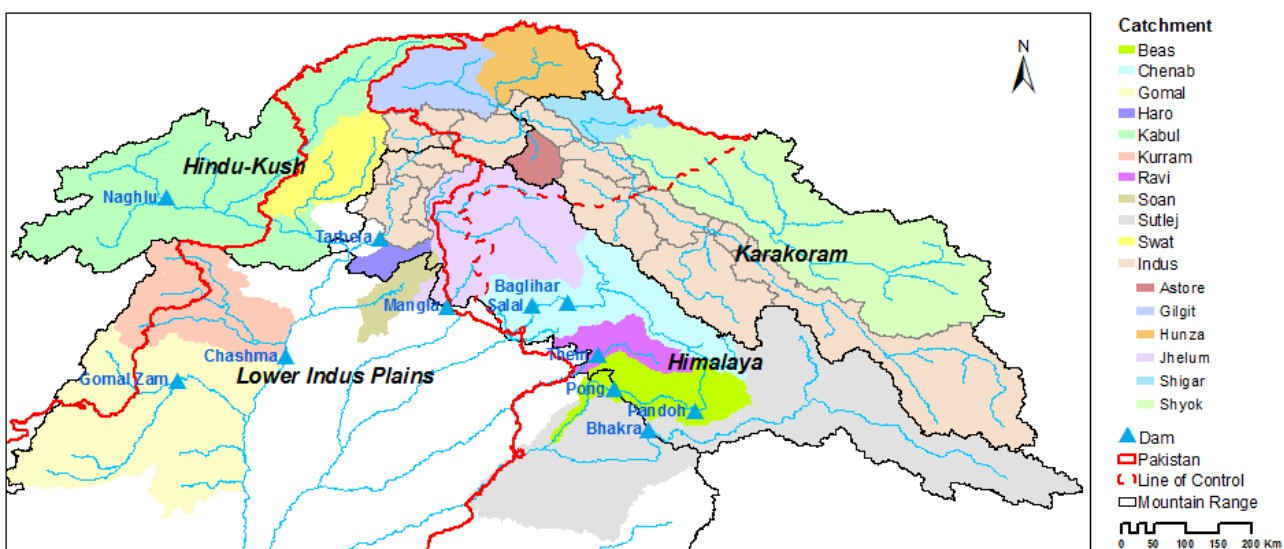

**Figure 1: Map of UIB showing sub-basins and location of major dams**

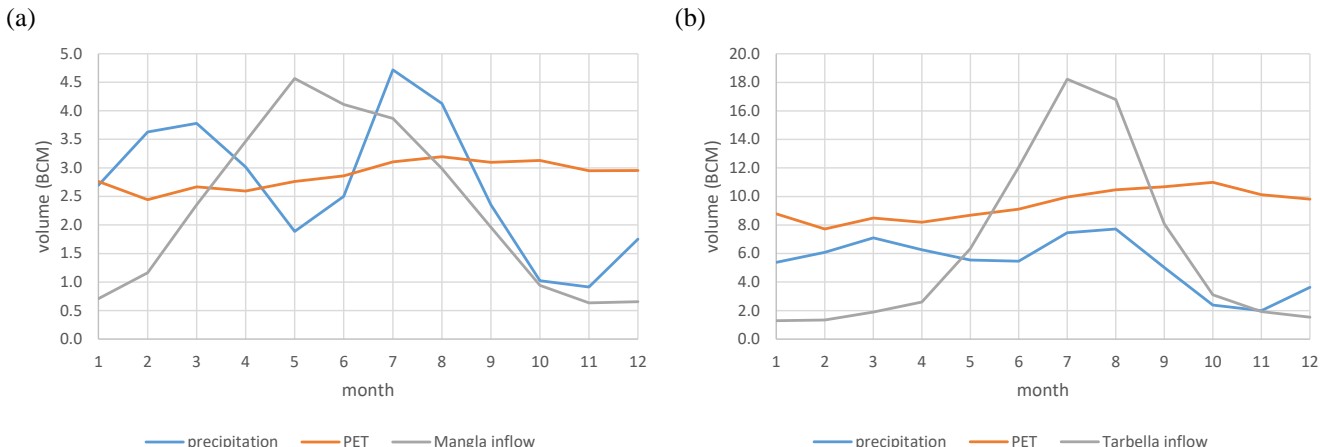

**Figure 2: Annual cycle of mean precipitation, PET and inflow for (a) Jhelum at Mangla and (b) Indus at Tarbella (note different volume scales).**

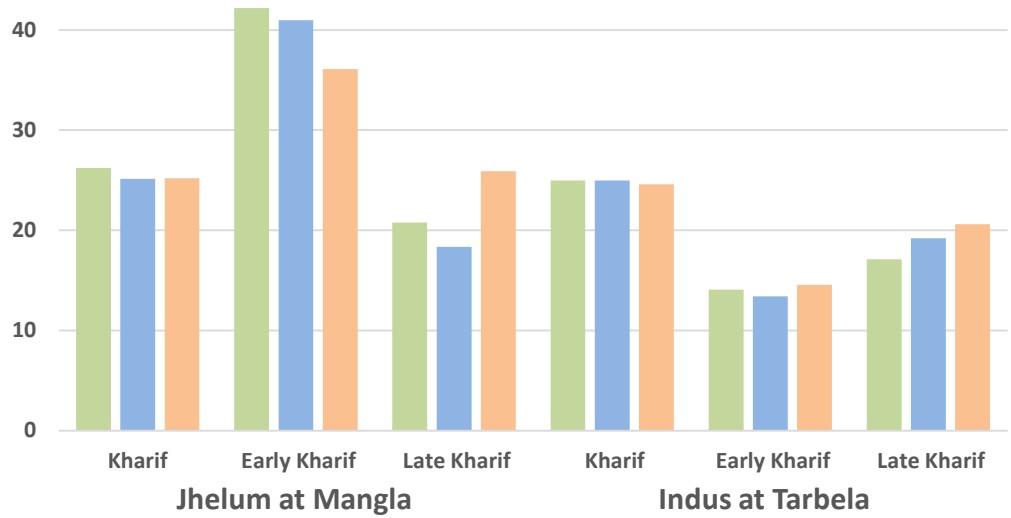

**Figure 3: BJP cross-validated skill scores, % skill gain relative to climatology, for CRPS (green), RMSE (blue) and IQR (orange) skill scores. Less than 5 is considered to be a forecast with very low skill. Between 5 and 15 is considered low skill. Between 15 and 30 is considered moderate skill, and higher than 30 is considered to be a forecast with high skill.**

(a)

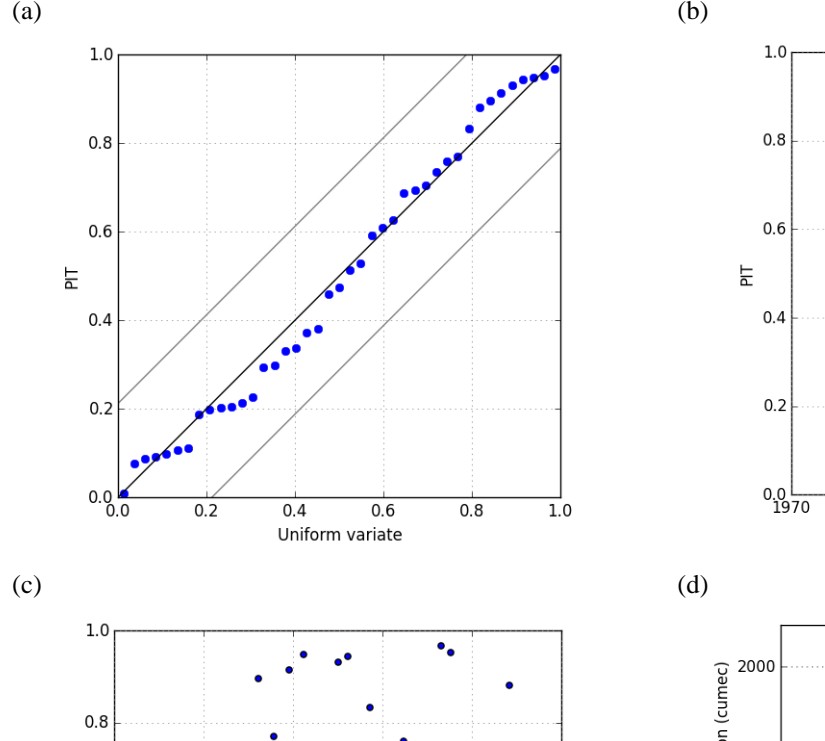

(b)

(c)

(d)

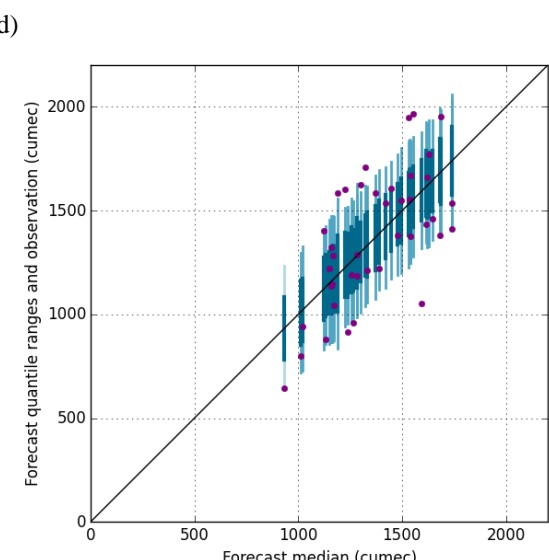

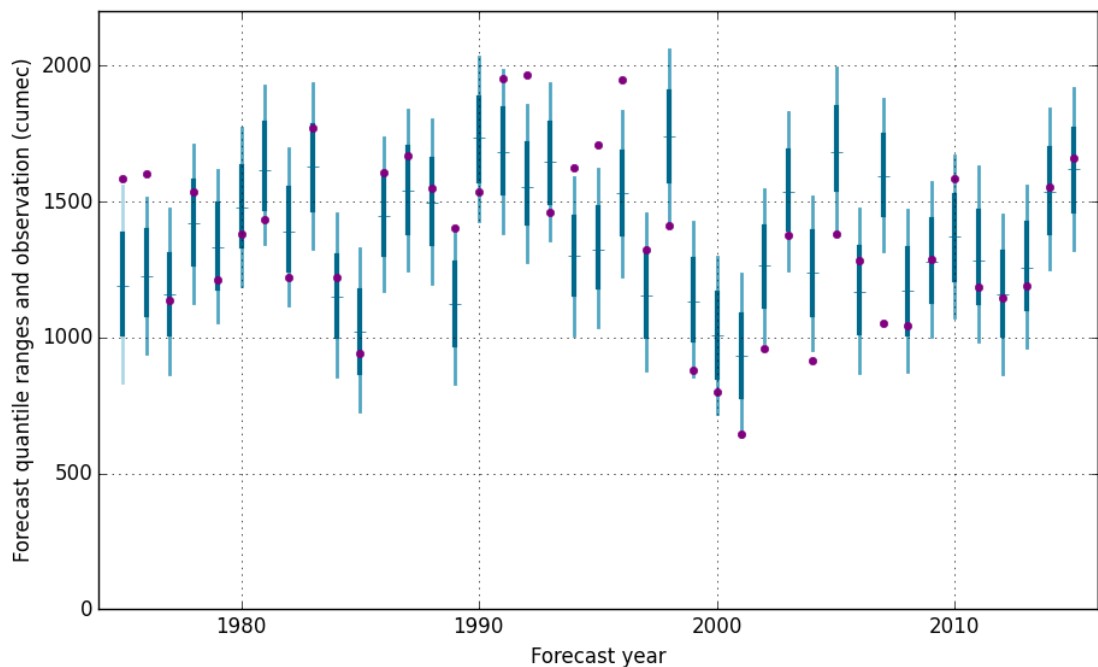

**Figure 4: BJP cross-validated forecasts for Jhelum at Mangla Kharif for 1975-2015; (a) PIT uniform probability plot (1:1 black line, theoretical uniform distribution; grey lines, Kolmogorov 5% significance bands; blue points, PIT values of forecast streamflow); (b) chronological PIT plot; (c) median PIT plot; (d) forecast quantiles and observed plotted according to forecast median (1:1 line, forecast median; dark vertical line, forecast [0.25, 0.75] quantile range; light and dark vertical line, forecast [0.10, 0.90] quantile range; dots, observed inflow); (e) chronological forecast quantile range and observations (dark blue, forecast [0.25, 0.75]; light and dark blue, forecast [0.10, 0.90]; crosses, forecast median; dots, observed).**

(a)

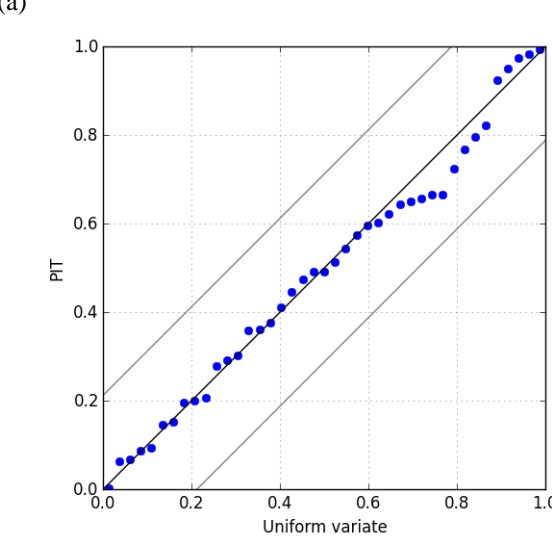

(b)

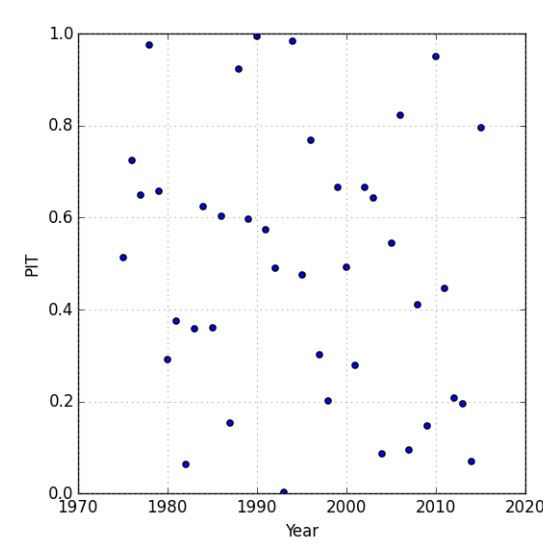

(c)

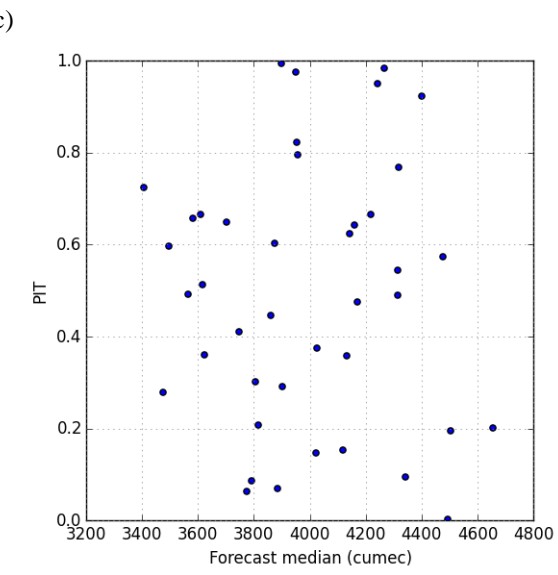

(d)

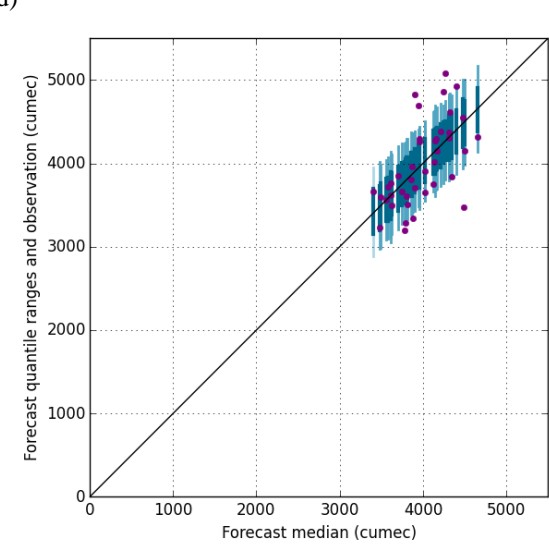

(e)

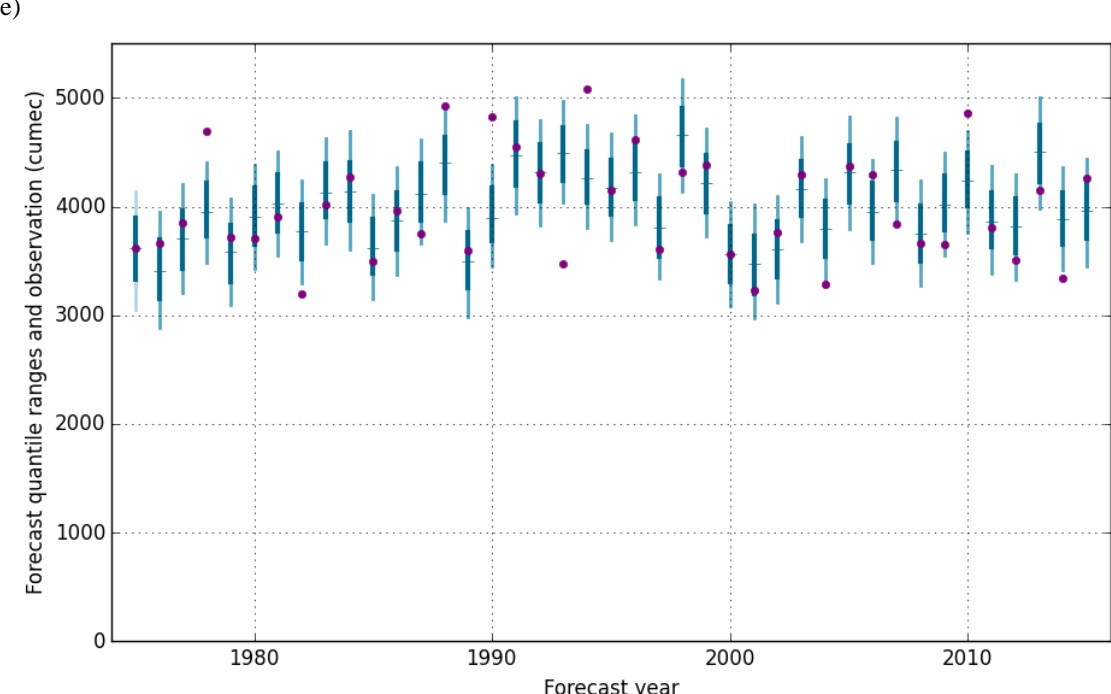

**Figure 5: as in Figure 4 for BJP cross-validated forecasts for Indus at Tarbela Kharif for 1975-2015**

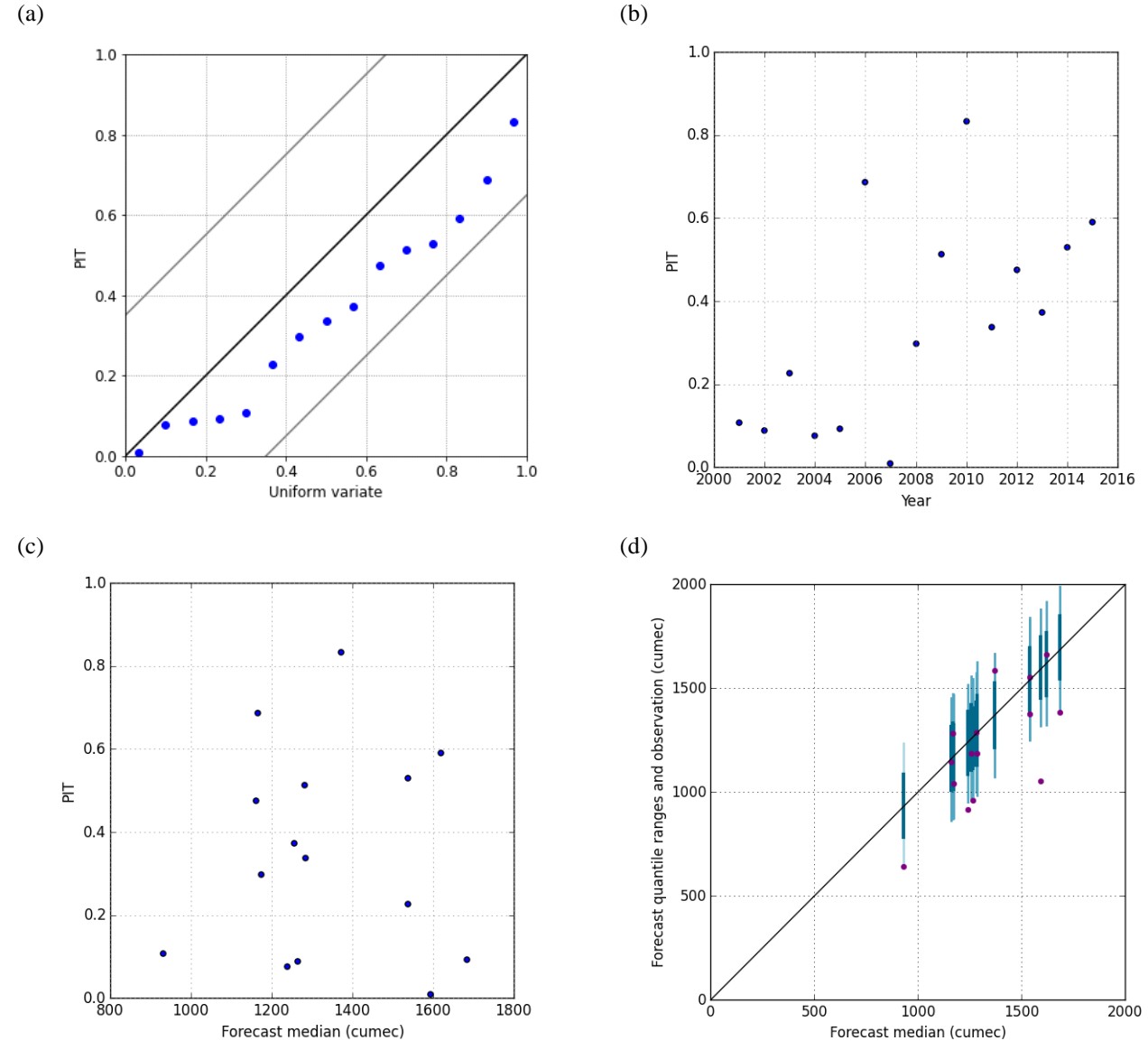

**Figure 6: as in Figure 4 for BJP cross-validated forecasts for Jhelum at Mangla Kharif for SRM period of 2001-2015, except for (e) see 2001-2015 period of Figure 4 (e).**

(a)

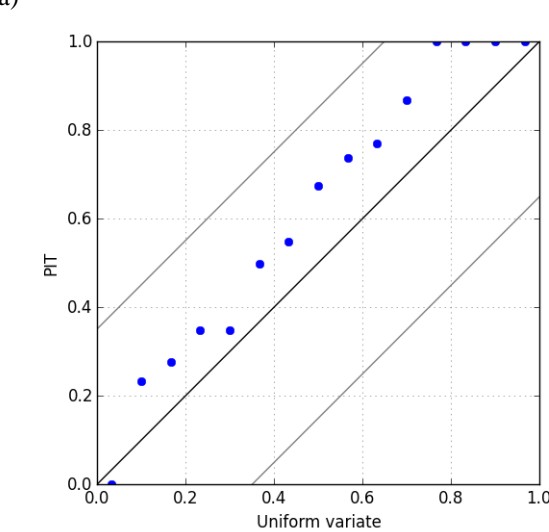

(b)

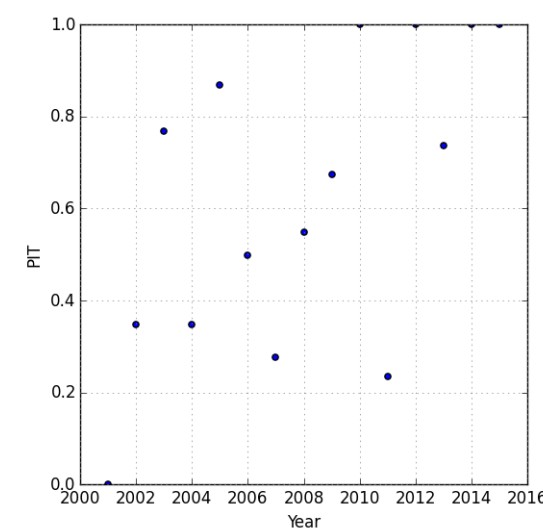

(c)

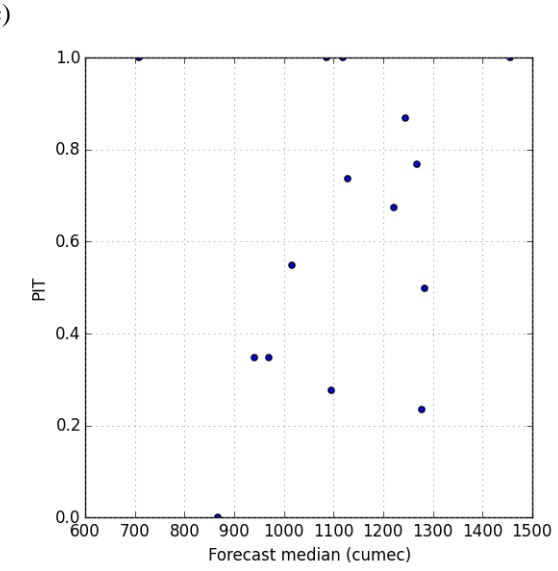

(d)

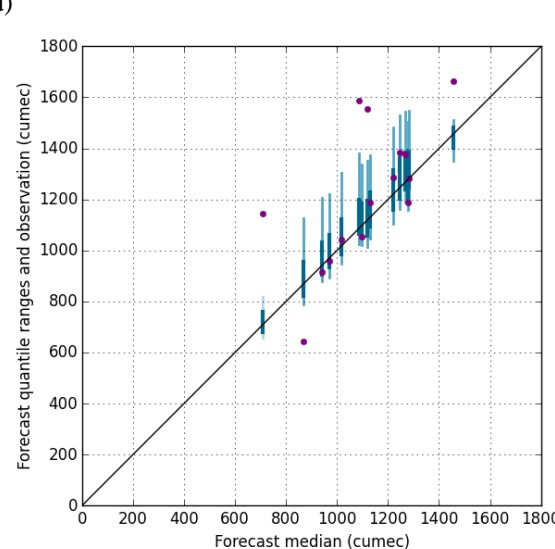

(e)

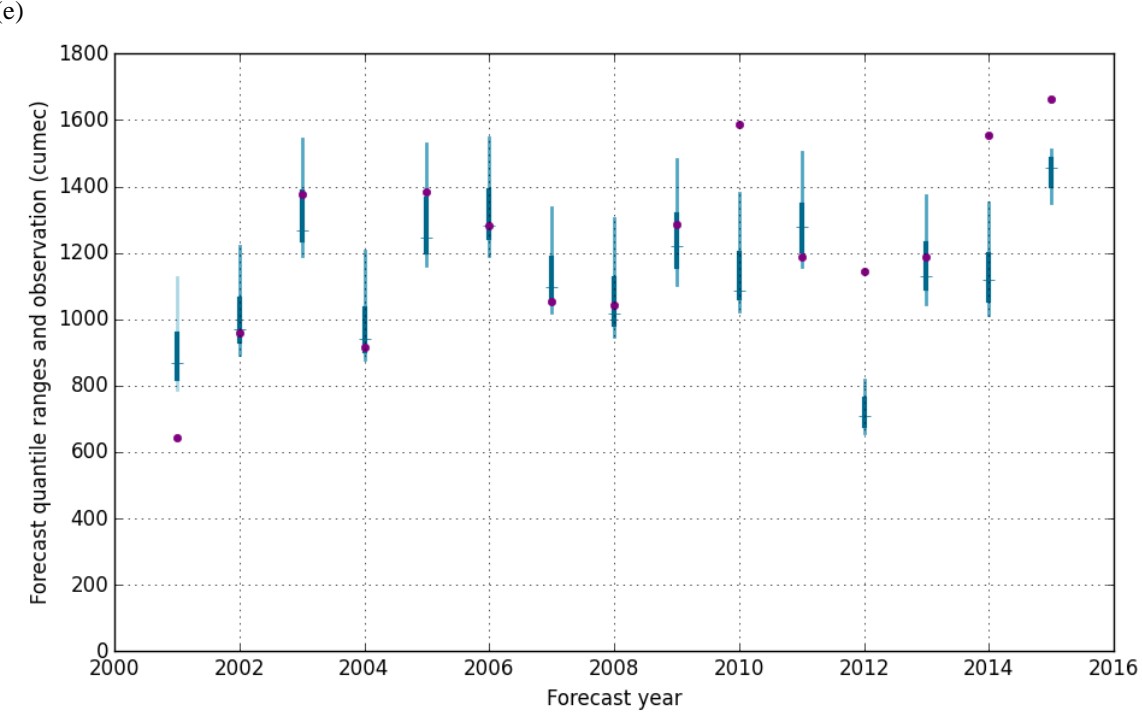

**Figure 7: as in Figure 4 for SRM ESP forecasts for Jhelum at Mangla Kharif for SRM period of 2001-2015**