# Peer review of "Seasonal streamflow forecasting in the Upper Indus Basin of Pakistan: an assessment of methods"

_Hydrology and Earth System Sciences, 2017_

## Referee Comment (RC1) · Anonymous Referee #1 · 2 Nov 2017

In this study performance of streamflow forecasts for Kharif Season (April-September) in the Upper Indus Basin of Pakistan is assessed. Streamflow forecasts are generated using the Bayesian joint probability (BJP) approach. Several predictors such as antecedent flow, climate indicators, and ESP based streamflow forecasts are used to test the performance of the streamflow forecasts. The study finds that in general BJP streamflow forecasts based on predictors antecedent flow and climate indicators perform the best. Variation in the skill is found for the focus basins, and for the early and late part of the season. In general, the manuscript is well organized and methods are technically sound. I do have a few comments/suggestions, some of which are moderate to major, which need to be addressed before publication.

[Figure]

Major comments:

(1) It would be helpful, mostly for the readers who are not well aware of the seasonal cycle of climate in the region, to add a figure for both basins that show the seasonal cycle of precipitation, temperature, runoff/streamflow. Similar to Fig. 2 of this manuscript http://journals.ametsoc.org/doi/full/10.1175/JHM-D-14-0213.1. Such a figure would provide a needed background to the readers about the region and also help interpret the results of streamflow forecasts evaluation. (2) The authors mention lack of climate forecasts skill in this region. I would encourage them to show a map(s) of the long-term skill of at least rainfall (winter) and temperature (winter and summer) in the region. I think a case for using statistical forecasts such as ones presented in this study can be made better if statistical forecast skill is demonstrated relative to the skill of dynamical forecasts, not just climatological forecasts. As of now, there are several global dynamical forecasts systems that provide operational seasonal forecasts. One of them being the North American Multimodel Ensemble (NMME, http://www.cpc.ncep.noaa.gov/products/NMME/). (3) The authors use March streamflow is the only predictors reflecting antecedent conditions, it is not clear why other variables such as snow water equivalent, soil moisture, total water storage were not used. Nowadays observations (through remote sensing) or simulations (e.g. through GLDAS https://ldas.gsfc.nasa.gov/index.php) of those variables are readily available. Especially in a region where snowmelt runoff is dominant, I would think snow and soil moisture would provide some streamflow forecast skill. (4) I would also encourage the authors to provide some more details regarding the PIT plots in the method section. To my knowledge PIT is not a typical metric used for forecast evaluation so it would help the readers to get a bit more details on them and also briefly describe what each type of the figures (a through e) highlights regarding the forecast skill.

Minor comments:

(5) P2, L24: Not only P and T but other atmospheric forcings as well. (6) P2, L29: This statement regarding the skill of dynamical forecast skill should be made more specific,

e.g. mention the regions and seasons etc. (7) P3, L23: Summer streamflow would depend upon winter T too, as winter T would influence snow accumulation. Please revise. (8) P4, L5-10: These sentences are confusing and hard to understand. (9) P5, L21: Please see comment #2. (10) Results in Table 1 and 2: It is not clear if those results are after cross-validation or before? Or are the results presented in Figure 3 onward are cross-validated? I would suggest comparing the cross-validated skill vs the skill calculated using the entire period. (11) Section 3.3: Suggest dividing this section into three sub-sections to discuss each of the verification scores separately. (12) P8, L2: It is surprising to see that MEI (May-June) from the previous year is a skillful predictor. Could you comment on why that may be? During May-June, ENSO events are in initial development stage and sometimes may change signs in the later part of the year so it is surprising that in this case, you are finding MEI May-June to be a skillful predictor for the streamflow of the following year. (13) P8, L14-15. I thought that in some cases March flow was the highest skill predictor and adding any predictor didn't increase the skill so why are you using both March flow and climate predictors here? (14) Figure 5 and 6. These figures are used to compare the skill of BJP vs SRM based streamflow forecasts. I think it would be better to combine them both into one figure. Maybe just show SRM forecasts with a different color. (15) Conclusion: The last two bullet points are not really findings. I suggest to separately discuss them after listing the findings. Also please mention here the current state-of-the-practice for generating streamflow forecasts in the region and the value the methods explored in this study will add.

---

## Referee Comment (RC2) · Anonymous Referee #2 · 14 Dec 2017

The authors have assessed three different methods - Bayesian Joint Probability (BJP), the Snowmelt Runoff Model (SRM) and a hybrid approach (SRM - Ensemble Streamflow Prediction inflow means as additional predictor in BJP approach) for forecasting seasonal streamflow to the two largest dams in the Upper Indus Basin, Pakistan. The authors concluded that BJP approach is simple and it worked well to provide probabilistic seasonal streamflow forecasts.

The topic is relevant for publication in HESS. Overall, the paper is well written. I recommend a moderate revision to the manuscript and the following concerns need to be addressed:

[Figure]

Major Concerns: 1. Under the BJP approach, was the conditional multivariate normal distributions fit over the entire season or on monthly basis? How many samples were generated through Monte Carlo Simulations under the BJP approach? Provide details. 2. Page 7, lines 24-27, the skill of using March flow and/or one climate predictor looks very similar to each other. The authors are recommended to use statistical significance test to compare if the skills are significantly different from each other. 3. Given that most of the streamflow at Indus River at Tarbela is snowmelt driven, use of a direct or indirect indicator of snow as one of the predictors, along with the projected summer air temperature can improve the forecasting skill. The authors are encouraged to consider global precipitation (for winter) and air temperature forecasts as predictors, which can represent snow as one of the inputs to the model. 4. It is not clear why MEI for May and Jun from previous year enhanced the skill score for Indus at Tarbela? Explain. 5. Page 7, line 1, how good or better the skill enhancement is if SSCRSP (or SSRMSE) changes from 21 to 24.3 (within moderate skill range in Table 1)? Does it reduce uncertainty? Clarify. 6. In Table 3, it will be good to know the correlations that are statistically significant (e.g. at 95% confidence interval) based on the sample size. 7. Page 10, lines 2-7, the hypotheses listed are not clear. As mentioned by the authors earlier, it is already known the snowmelt plays an important role for Indus River at Tarbela. So it not a hypothesis. Also, the results indicated that adding NAO, when used as a predictor, did not improve forecasting skill.

Minor Concerns: 8. Did the models use monthly (or daily) data for the model fitting? If so, it needs to be clearly stated. 9. Page 6, lines 27 – 30, RMSEP needs to be used instead of RMSE. Also RMSEP needs to be defined in the text. 10. In figures 3a, 4a, 5a and 6a, what are the bounding lines (is it 95% Confidence Interval)?

---

## Author Comment (AC1) · 13 Feb 2018

**Response to interactive comment by Anonymous Referee #1**

In this study performance of streamflow forecasts for Kharif Season (April-September) in the Upper Indus Basin of Pakistan is assessed. Streamflow forecasts are generated using the Bayesian joint probability (BJP) approach. Several predictors such as antecedent flow, climate indicators, and ESP based streamflow forecasts are used to test the performance of the streamflow forecasts. The study finds that in general BJP streamflow forecasts based on predictors antecedent flow and climate indicators perform the best. Variation in the skill is found for the focus basins, and for the early and late part of the season. In general, the manuscript is well organized and methods are technically sound. I do have a few comments/suggestions, some of which are moderate to major, which need to be addressed before publication.

**Response:** Thank you for your encouraging and helpful review.

Major comments:

(1) It would be helpful, mostly for the readers who are not well aware of the seasonal cycle of climate in the region, to add a figure for both basins that show the seasonal cycle of precipitation, temperature, runoff/streamflow. Similar to Fig. 2 of this manuscript http://journals.ametsoc.org/doi/full/10.1175/JHM-D-14-0213.1. Such a figure would provide a needed background to the readers about the region and also help interpret the results of streamflow forecasts evaluation.

**Response:** We will add a new figure to show the seasonal cycle of streamflow, precipitation and calculated potential evapotranspiration for our two study basins.

(2) The authors mention lack of climate forecasts skill in this region. I would encourage them to show a map(s) of the long-term skill of at least rainfall (winter) and temperature (winter and summer) in the region. I think a case for using statistical forecasts such as ones presented in this study can be made better if statistical forecast skill is demonstrated relative to the skill of dynamical forecasts, not just climatological forecasts. As of now, there are several global dynamical forecasts systems that provide operational seasonal forecasts. One of them being the North American Multimodel Ensemble (NMME, http://www.cpc.ncep.noaa.gov/products/NMME/).

**Response:** We agree with the comment that it will be useful to assess statistical forecast skill together with dynamical forecast skill. The reason that this has not been done in this study has do with the strong practical application focus of the tool development. Because of the need to model glacier and snow processes, sophisticated hydrological models that could ingest climate forecasts are of limited availability. Furthermore, the region does not have ready access to real-time seasonal climate forecasts, although this problem can be overcome. The purpose of this study is therefore to develop tools that can be easily implemented based on infrastructure that exists today. As infrastructure (like hydrological models) improves, an extended study on a fully dynamical forecasting system will be highly appropriate. As dynamical forecasts need to be post-processed for use in hydrological forecasting, the linkage of dynamical climate model forecasts with hydrological models in the region would require substantial additional research.

In the Introduction, we will provide more information on the background of this study and clarify the strong practical application focus of this study, including why a fully dynamical forecasting system has not been investigated. We will also add to the discussion, regarding potential further research with dynamical models.

(3) The authors use March streamflow is the only predictors reflecting antecedent conditions, it is not clear why other variables such as snow water equivalent, soil moisture, total water storage were not used. Nowadays observations (through remote sensing) or simulations (e.g. through GLDAS https://ldas.gsfc.nasa.gov/index.php) of those variables are readily available. Especially in a region where snowmelt runoff is dominant, I would think snow and soil moisture would provide some streamflow forecast skill.

**Response:** We did investigate a MODIS-based snow cover product (post-processed to remove cloud cover effects) and did find a relatively high correlation with streamflow. For example, for the Jhelum catchment the MODIS snow cover at the end of March has a 0.68 correlation with Kharif season streamflow. However, the MODIS data only commenced 2000 and so this limited availability of data resulted in its exclusion from the final set of model evaluations. As March flow correlation with Kharif season streamflow is also 0.68, a snow cover predictor is not expected to be a demonstrably better predictor than March flow in this case.

(4) I would also encourage the authors to provide some more details regarding the PIT plots in the method section. To my knowledge PIT is not a typical metric used for forecast evaluation so it would help the readers to get a bit more details on them and also briefly describe what each type of the figures (a through e) highlights regarding the forecast skill.

**Response:** We will add a more detailed explanation, as well as further cite published literature regarding the use of PIT plots to evaluate probabilistic forecasts.

Minor comments:

(5) P2, L24: Not only P and T but other atmospheric forcings as well.

**Response:** We will change the text to remove specific mention of P and T: "*Dynamical approaches use hydrological models initialised with observed inputs up to the beginning of the forecast season to account for antecedent conditions, that can be driven either by historical or modelled climate inputs.*"

(6) P2, L29: This statement regarding the skill of dynamical forecast skill should be made more specific, e.g. mention the regions and seasons etc.

**Response:** We will add a summary, as a specific example, of a published evaluation of dynamical climate forecast skill for the Asian summer monsoon from ECMWF and NCEP forecast systems.

(7) P3, L23: Summer streamflow would depend upon winter T too, as winter T would influence snow accumulation. Please revise.

**Response:** We will revise to include reference to winter T: "*The predominant source of flow in the UIB is snowmelt, with glacier melt a secondary source, with 80% of flow occurring during the June-September summer period. Interannual flow variability is thus controlled by two processes, snow accumulation as determined by winter precipitation and temperature and meltwater generation as determined by summer temperatures. Hence snowmelt-generated flow is a function of winter precipitation and temperature and also summer temperature, whereas glacier melt is primarily a function of summer temperature, although glacier melt rates are also influenced by snow cover.*"

(8) P4, L5-10: These sentences are confusing and hard to understand.

**Response:** These sentences will be replaced with a clearer statement: "*Useful climate indices should relate to the weather prior to the forecast season, providing an indication of snow accumulation, and*

*also to the weather within the forecast season, influencing temperature and hence snow and glacier melt rates.*"

(9) P5, L21: Please see comment #2.

**Response:** See response to comment #2 above.

(10) Results in Table 1 and 2: It is not clear if those results are after cross-validation or before? Or are the results presented in Figure 3 onward are cross-validated? I would suggest comparing the cross-validated skill vs the skill calculated using the entire period.

**Response:** These are cross-validated results, and the Table's headings will be updated to reflect this. In the paper we will re-emphasis our case that cross validated results are more representative of the real skill and reduce the chances of overfitting. Hence we do not want to show results without cross-validation given statistical methods are prone to artificial skill and overfitting.

(11) Section 3.3: Suggest dividing this section into three sub-sections to discuss each of the verification scores separately.

**Response:** As we rely on cited references for the provision of detailed descriptions of the skills scores, we feel there is not enough stand-alone material for each to justify its own sub-section.

(12) P8, L2: It is surprising to see that MEI (May-June) from the previous year is a skillful predictor. Could you comment on why that may be? During May-June, ENSO events are in initial development stage and sometimes may change signs in the later part of the year so it is surprising that in this case, you are finding MEI May-June to be a skillful predictor for the streamflow of the following year.

**Response:** The MEI (May-June) predictor skill relates to autumn/winter snow accumulation, a lag of 4 months to snow accumulation from October onwards. It is thus not unreasonable that circulation systems bringing moisture into the region during autumn/winter are influenced by the forcing initiated by ENSO processes during the summer.

There are many supporting references presenting details of ENSO/precipitation teleconnections for the region, with several cited in the manuscript. As an example, Mariotti (2007)[1] notes: "*The associated circulation pattern during El Nino (La Nina) involves a southwesterly (northeasterly) moisture flux that brings more (less) moisture into this region* [southwest central Asia]. *This flux flows along the northwestern flank of the large-scale high pressure anomaly over the Indian and western Pacific Oceans, broadly, the western pole of the Southern Oscillation see-saw pattern. Unlike many ENSO teleconnections in which lower pressure and weaker subsidence leads to more precipitation, this mechanism does not require a change in the local pressure, but rather involves a change in the tropical moisture supply to subtropical-midlatitude storms.*"

 (13) P8, L14-15. I thought that in some cases March flow was the highest skill predictor and adding any predictor didn't increase the skill so why are you using both March flow and climate predictors here?

**Response:** For Indus at Tarbela (Table 2), inclusion of a climate predictor in addition to the March flow predictor increased skill in all cases. For Jhelum at Mangla, there were mixed results as adding the climate predictor slightly decreased the skill for the full 1975-2015 period but increased the skill
* * *
[1] Mariotti, A.: How ENSO impacts precipitation in southwest central Asia, Geophysical Research Letters, 34, 10.1029/2007GL030078, 2007.

for the 2001-2015 period. As the 2001-2015 period is the comparison period for SRM, we think it is acceptable to use these predictor combinations.

(14) Figure 5 and 6. These figures are used to compare the skill of BJP vs SRM based streamflow forecasts. I think it would be better to combine them both into one figure. Maybe just show SRM forecasts with a different color.

**Response:** We are concerned that combining multiple forecast evaluations onto single figures would look cluttered in some circumstances, such as Panel (d), hence out preference is to maintain these figures as they are.

(15) Conclusion: The last two bullet points are not really findings. I suggest to separately discuss them after listing the findings. Also please mention here the current state-of-the-practice for generating streamflow forecasts in the region and the value the methods explored in this study will add.

**Response:** Agreed, these last two bullet points will be presented as a separate paragraph after the list of findings. We will add a description of current methods and add a discussion of the value added by our new approach.

---

## Author Comment (AC2) · 13 Feb 2018

**Response to interactive comment by Anonymous Referee #2**

The authors have assessed three different methods - Bayesian Joint Probability (BJP), the Snowmelt Runoff Model (SRM) and a hybrid approach (SRM - Ensemble Streamflow Prediction inflow means as additional predictor in BJP approach) for forecasting seasonal streamflow to the two largest dams in the Upper Indus Basin, Pakistan. The authors concluded that BJP approach is simple and it worked well to provide probabilistic seasonal streamflow forecasts. The topic is relevant for publication in HESS. Overall, the paper is well written. I recommend a moderate revision to the manuscript and the following concerns need to be addressed:

**Response:** Thank you for your encouraging and helpful review.

Major Concerns:

1. Under the BJP approach, was the conditional multivariate normal distributions fit over the entire season or on monthly basis? How many samples were generated through Monte Carlo Simulations under the BJP approach? Provide details.

**Response:** We will modify the text to include clarifying information: "*The cross-validated BJP forecast performance was assessed for 1975-2015 (41 seasons), with the BJP models calibrated on a seasonal basis (i.e. 40 data points) using 1000 MCMC samples for each of the leave-one-out calibrations.*"

2. Page 7, lines 24-27, the skill of using March flow and/or one climate predictor looks very similar to each other. The authors are recommended to use statistical significance test to compare if the skills are significantly different from each other.

**Response:** We will add uncertainty estimates to the skills scores, from bootstrapping, to aid interpretation of how different the forecast models are from one another.

3. Given that most of the streamflow at Indus River at Tarbela is snowmelt driven, use of a direct or indirect indicator of snow as one of the predictors, along with the projected summer air temperature can improve the forecasting skill. The authors are encouraged to consider global precipitation (for winter) and air temperature forecasts as predictors, which can represent snow as one of the inputs to the model.

**Response:** We agree that GCM forecasts of precipitation and temperature could potentially be used as predictors, however in this work we are assuming that the water resources practitioners do not readily have access to GCM seasonal climate forecast data (including hindcasts, needed for model establishment). Hence out approach relies on information regarding temperature and precipitation being captured by our selected climate index predictors (statistically). This could be the subject of future research with dynamical models, so we will mention this in our revised discussion.

4. It is not clear why MEI for May and Jun from previous year enhanced the skill score for Indus at Tarbela? Explain.

Response: The MEI (May-June) predictor skill relates to autumn/winter snow accumulation, a lag of 4 months to snow accumulation from October onwards. It is thus not unreasonable that circulation systems bringing moisture into the region during autumn/winter are influenced by the forcing initiated by ENSO processes during the summer.

There are many supporting references presenting details of ENSO/precipitation teleconnections for the region, with several cited in the manuscript. As an example, Mariotti (2007)[2] notes: "*The associated circulation pattern during El Nino (La Nina) involves a southwesterly (northeasterly) moisture flux that brings more (less) moisture into this region* [southwest central Asia]. *This flux flows along the northwestern flank of the large-scale high pressure anomaly over the Indian and western Pacific Oceans, broadly, the western pole of the Southern Oscillation see-saw pattern. Unlike many ENSO teleconnections in which lower pressure and weaker subsidence leads to more precipitation, this mechanism does not require a change in the local pressure, but rather involves a change in the tropical moisture supply to subtropical-midlatitude storms.*"

5. Page 7, line 1, how good or better the skill enhancement is if SSCRSP (or SSRMSE) changes from 21 to 24.3 (within moderate skill range in Table 1)? Does it reduce uncertainty? Clarify.

**Response:** As described in the verification methods section, improvements in CRPS reflect improvement in accuracy and/or sharpness and improvements in RMSE reflect improvements in accuracy of the median only. So an inference can be made through comparative analysis of the various skill metrics, including the IQR. We acknowledge that it is a small difference and will add uncertainty via bootstrap results to help clarify this.

6. In Table 3, it will be good to know the correlations that are statistically significant (e.g. at 95% confidence interval) based on the sample size.

**Response:** We will indicate statistical significance in the table.

7. Page 10, lines 2-7, the hypotheses listed are not clear. As mentioned by the authors earlier, it is already known the snowmelt plays an important role for Indus River at Tarbela. So it not a hypothesis. Also, the results indicated that adding NAO, when used as a predictor, did not improve forecasting skill.

**Response:** We will re-word to avoid the confusion caused by the term "hypothesised". Table 2 shows that the NAO predictor did add some skill, however not as much as the selected ENSO based predictor.

Minor Concerns:

8. Did the models use monthly (or daily) data for the model fitting? If so, it needs to be clearly stated.

**Response:** The BJP is calibrated to seasonal data (i.e. 41 data points 1975 to 2015). This will be clarified in the text.

RC2:9. Page 6, lines 27 – 30, RMSEP needs to be used instead of RMSE. Also RMSEP needs to be defined in the text.

**Response:** We will correct and add a citation that includes the derivation of RMSEP (Wang, Q. J., and Robertson, D. E.: Multisite probabilistic forecasting of seasonal flows for streams with zero value occurrences, Water Resources Research, 47, 10.1029/2010WR009333, 2011.)

RC2:10. In figures 3a, 4a, 5a and 6a, what are the bounding lines (is it 95% Confidence Interval)?
* * *
[2] Mariotti, A.: How ENSO impacts precipitation in southwest central Asia, Geophysical Research Letters, 34, 10.1029/2007GL030078, 2007.

Response: As stated in the figure caption, these are Kolmogorov 5% significance bands. We will clarify this in our revised text referring to these figures.

---

## Author Response (AR1)

**Response to interactive comment by Anonymous Referee #1**

In this study performance of streamflow forecasts for Kharif Season (April-September) in the Upper Indus Basin of Pakistan is assessed. Streamflow forecasts are generated using the Bayesian joint probability (BJP) approach. Several predictors such as antecedent flow, climate indicators, and ESP based streamflow forecasts are used to test the performance of the streamflow forecasts. The study finds that in general BJP streamflow forecasts based on predictors antecedent flow and climate indicators perform the best. Variation in the skill is found for the focus basins, and for the early and late part of the season. In general, the manuscript is well organized and methods are technically sound. I do have a few comments/suggestions, some of which are moderate to major, which need to be addressed before publication.

**Response:** Thank you for your encouraging and helpful review.

Major comments:

(1) It would be helpful, mostly for the readers who are not well aware of the seasonal cycle of climate in the region, to add a figure for both basins that show the seasonal cycle of precipitation, temperature, runoff/streamflow. Similar to Fig. 2 of this manuscript http://journals.ametsoc.org/doi/full/10.1175/JHM-D-14-0213.1. Such a figure would provide a needed background to the readers about the region and also help interpret the results of streamflow forecasts evaluation.

**Response:** We agree and have added Figure 2 to show the seasonal cycle of streamflow, precipitation and calculated potential evapotranspiration for our two study basins.

(2) The authors mention lack of climate forecasts skill in this region. I would encourage them to show a map(s) of the long-term skill of at least rainfall (winter) and temperature (winter and summer) in the region. I think a case for using statistical forecasts such as ones presented in this study can be made better if statistical forecast skill is demonstrated relative to the skill of dynamical forecasts, not just climatological forecasts. As of now, there are several global dynamical forecasts systems that provide operational seasonal forecasts. One of them being the North American Multimodel Ensemble (NMME, http://www.cpc.ncep.noaa.gov/products/NMME/).

**Response:** We agree with the comment that it can be useful to assess statistical forecast skill together with dynamical forecast skill. The reason that this has not been done in this study has do with the strong practical application focus of the tool development. Because of the need to model glacier and snow processes, sophisticated hydrological models that could ingest climate forecasts are of limited availability. Furthermore, the region does not have ready access to real-time seasonal climate forecasts, although this problem can be overcome. The purpose of this study is therefore to develop tools that can be easily implemented based on infrastructure that exists today. As infrastructure (like hydrological models) improves, an extended study on a fully dynamical forecasting system will be highly appropriate. As dynamical forecasts need to be post-processed for use in hydrological forecasting, the linkage of dynamical climate model forecasts with hydrological models in the region would require substantial additional research.

Additionally, a recently published assessment of forecast skill over Pakistan and Afghanistan for NMME May 1 hindcasts (for May to November) concluded that the MMEM, that generally exceeded the skill of any individual model, provides little benefit over climatology (Cash, B. A., Manganello, J. V., and Kinter, J. L.: Evaluation of NMME temperature and precipitation bias and forecast skill for South Asia, Clim Dyn, 10.1007/s00382-017-3841-4, 2017). Given this assessment, we would not

expect NMME forecast precipitation or temperature for our study region to add skill as additional predictors to the BJP. We have thus added the following text to the Introduction:

"*Cash et al. (2017) assessed monthly North American Multi-Model Ensemble (Kirtman et al., 2013) hindcasts initialised May 1 for May to November for South Asia, including the mountainous areas of Afghanistan and Pakistan. They concluded that the multi-model ensemble mean temperature and precipitation forecasts, while generally exceeding the skill of any individual model, provided little benefit over climatology.*"

In the Introduction, we have provided more information on the background of this study and clarify the strong practical application focus of this study, including why a fully dynamical forecasting system has not been investigated. We have also added to the discussion, regarding potential further research with dynamical models.

(3) The authors use March streamflow is the only predictors reflecting antecedent conditions, it is not clear why other variables such as snow water equivalent, soil moisture, total water storage were not used. Nowadays observations (through remote sensing) or simulations (e.g. through GLDAS https://ldas.gsfc.nasa.gov/index.php) of those variables are readily available. Especially in a region where snowmelt runoff is dominant, I would think snow and soil moisture would provide some streamflow forecast skill.

**Response:** We did investigate a MODIS-based snow cover product (post-processed to remove cloud cover effects) and did find a relatively high correlation with streamflow. For example, for the Jhelum catchment the MODIS snow cover at the end of March has a 0.68 correlation with Kharif season streamflow. However, the MODIS data only commenced 2000 and so this limited availability of data resulted in its exclusion from the final set of model evaluations. As March flow correlation with Kharif season streamflow is also 0.68, a snow cover predictor is not expected to be a demonstrably better predictor than March flow in this case.

In response to the Editor's additional recommendations, we undertook analysis of GLDAS-SWE hindcast performance for our two study basins. Firstly there is the issue of record length, as GLDAS-2.0 only covers to 2010 (i.e. it isn't up to date and thus unsuitable for real-time operational forecasting) and GLDAS-2.1 only commences in 2000, i.e. a shorter period than the predictors assessed in the manuscript. Such a short period was also our justification for not using MODIS snow cover as a predictor, which similarly commenced in 2000. Despite this short record length, we have assessed 2000-2015 annual correlation of v2.1 SWEMarch with Kharif (April-September) flow and QMarch for our two study basins.

- For Jhelum at Mangla, SWEMarch and QMarch correlation with Kharif flow are 0.50 and 0.73, respectively. Also, there is a 0.87 correlation between SWEMarch and QMarch, suggesting SWEMarch doesn't provide additional information and hence skill above that provided by QMarch.
- For Indus at Tarbella, SWEMarch and QMarch correlation with Kharif flow are comparable at 0.56 and 0.55, respectively. The correlation between SWEMarch and QMarch is 0.77, suggesting they are not independent predictors.

Therefore, the limitation of short record length, lack of higher correlation with flow than that of the QMarch predictor, and relatively high cross-correlation with QMarch, leads us to conclude that GLDAS-SWE would not provide additional skill as a predictor for the BJP. We have added the following to section 4.1 Skills score to address this:

*"MODIS (Hall et al., 2010) snow-cover area and GLDAS-2.1 (Rodell et al., 2004) snow-water equivalent, additional measures of antecedent conditions, were also assessed as potential predictors. A significant limitation is the shorter record lengths for MODIS and GLDAS, as available data for both start in 2000. This is of particular concern for the BJP's leave-one-out cross-validation, as using short records to identify dynamical mechanisms is susceptible to spurious skill. Correlation analysis, cognisant of the short 2000-15 period, found that these snow products have a similar or lower correlation with Kharif flow (QKharif) compared to March flow (QMarch), and are relatively highly correlated with QMarch. Thus the limitation of short record length, lack of higher correlation with flow than that of the QMarch predictor, and relatively high cross-correlation with QMarch, leads us to conclude that they would not be expected to provide additional skill as a predictor for the BJP."*

(4) I would also encourage the authors to provide some more details regarding the PIT plots in the method section. To my knowledge PIT is not a typical metric used for forecast evaluation so it would help the readers to get a bit more details on them and also briefly describe what each type of the figures (a through e) highlights regarding the forecast skill.

**Response:** We agree and have extended the explanation of the use of PIT plots to evaluate probabilistic forecasts as follows (new text in **bold**).

*"Reliability refers to the statistical similarity between the forecast probabilities and the relative frequencies of events in the observations, which can be verified using probability integral transforms (PITs). The PIT represents the non-exceedance probability of observed streamflow obtained from the CDF of the ensemble forecast. If the forecast ensemble spread is appropriate and free of bias then observations will be contained within the forecast ensemble spread, with reliable forecasts having PIT values that follow a uniform distribution between 0 and 1 (Laio and Tamea, 2007). **Thus PIT plots are an efficient diagnostic to visually evaluate whether the forecast probability distributions are too wide or too narrow or are biased (under or over estimating) in their prediction of the observed distribution (Wang and Robertson, 2011). As outlined by (Thyer et al., 2009), PIT plot points falling on the 1:1 line indicate that the predicted distribution is a perfect match to the observed; observed PIT values of 0.0 or 1.0 indicate the corresponding observed data falls outside the predicted range, hence the predictive uncertainty is significantly underestimated; PIT values clustered around the midrange (i.e. a low slope in the 0.4 -0.6 uniform variate range) indicate the predictive uncertainty is overestimated; PIT values clustered around the tails (i.e. a high slope in the 0.4 -0.6 uniform variate range) indicate the predictive uncertainty is underestimated; and if PIT values at the theoretical median are higher than those of the uniform variate the predictions have an underprediction bias, and vice versa if they are lower than the uniform variate then the predictions have an overprediction bias**."*

Minor comments:

(5) P2, L24: Not only P and T but other atmospheric forcings as well.

**Response:** We have changed the text to remove specific mention of P and T, it now reads: *"Dynamical approaches use hydrological models initialised with observed inputs up to the beginning of the forecast season (to account for antecedent conditions) that can be driven either by historical or modelled climate inputs."*

(6) P2, L29: This statement regarding the skill of dynamical forecast skill should be made more specific, e.g. mention the regions and seasons etc.

**Response:** We have added a summary of published evaluations of dynamical climate forecast skill for the region, as follows:

*"Dynamical (i.e. climate model) forecasts of precipitation and temperature are often not sufficiently skilful in this region. For example, Kim et al. (2012) assessed retrospective seasonal forecasts of the Asian summer monsoon from ECMWF System 4 (Molteni et al., 2011) and NCEP CFSv2 (Saha et al., 2014), finding low skill for precipitation prediction and poor simulations of the Indian summer monsoon circulation. Cash et al. (2017) assessed monthly North American Multi-Model Ensemble (Kirtman et al., 2013) hindcasts initialised May 1 for May to November for South Asia, including the mountainous areas of Afghanistan and Pakistan. They concluded that the multi-model ensemble mean temperature and precipitation forecasts, while generally exceeding the skill of any individual model, provided little benefit over climatology."*

(7) P3, L23: Summer streamflow would depend upon winter T too, as winter T would influence snow accumulation. Please revise.

**Response:** We have revised to include reference to winter T, as follows:

*"The predominant source of flow in the UIB is snowmelt, with glacier melt a secondary source, with 80% of flow occurring during the June-September summer period. Interannual flow variability is thus controlled by two processes, snow accumulation as determined by winter precipitation and temperature and meltwater generation as determined by summer temperatures. Hence snowmelt-generated flow is a function of winter precipitation and temperature and also summer temperature, whereas glacier melt is primarily a function of summer temperature, although glacier melt is also influenced by snow cover (Charles, 2016)."*

(8) P4, L5-10: These sentences are confusing and hard to understand.

**Response:** These sentences have been replaced by the following sentence:

*"Useful climate indices should relate to the weather prior to the forecast season, providing an indication of snow accumulation, and also to the weather within the forecast season, influencing temperature and hence snow and glacier melt rates."*

(9) P5, L21: Please see comment #2.

**Response:** See response to comment #2 above.

(10) Results in Table 1 and 2: It is not clear if those results are after cross-validation or before? Or are the results presented in Figure 3 onward are cross-validated? I would suggest comparing the cross-validated skill vs the skill calculated using the entire period.

**Response:** These are cross-validated results, and the Table captions have been updated to reflect this. In the paper we will re-emphasis our case that cross validated results are more representative of the real skill and reduce the chances of overfitting. Hence we do not want to show results without cross-validation, given statistical methods are prone to artificial skill and overfitting.

(11) Section 3.3: Suggest dividing this section into three sub-sections to discuss each of the verification scores separately.

**Response:** As we rely on cited references for the provision of detailed descriptions of the skill scores, we feel there is not enough stand-alone material for each score to justify three sub-sections.

(12) P8, L2: It is surprising to see that MEI (May-June) from the previous year is a skillful predictor. Could you comment on why that may be? During May-June, ENSO events are in initial development stage and sometimes may change signs in the later part of the year so it is surprising that in this case, you are finding MEI May-June to be a skillful predictor for the streamflow of the following year.

**Response:** We hypothesised that the MEI (May-June) predictor skill relates to autumn/winter snow accumulation, a lag of 4 months to snow accumulation from October onwards. It is thus not unreasonable that circulation systems bringing moisture into the region during autumn/winter are influenced by the forcing initiated by ENSO processes during the summer. We investigated this further and have added the following text to the discussion:

"*For Mangla, the predictor combination that gave the best Kharif season cross-validated skill scores included an ENSO-based predictor ($SSI_{March}$) immediately before the season (Table 1), which makes sense intuitively as it represents a climate driver of both the snow accumulation before and precipitation conditions during the Kharif season. In contrast, for Tarbela a much earlier ENSO-based predictor ($MEI_{MayJun}$, i.e. May-Jun the year before) provides higher skill scores than the equivalent predictor immediately before the season ($MEI_{FebMar}$) (Table 2). To try to understand the dynamical mechanism by which $MEI_{MayJun}$ is providing skill in forecasting $Q_{Kharif}$, we compared MEI correlations with GLDAS $SWE_{March}$, $Q_{March}$ and $Q_{Kharif}$. Results were inconclusive, and perhaps impeded by the short record lengths given SWE is only available from 2000, as while $MEI_{MayJun}$ has a higher correlation with $Q_{Kharif}$ than $MEI_{FebMar}$ (0.76 versus 0.63, respectively) it has a slightly lower correlation with $SWE_{March}$ (0.48 versus 0.52, respectively). Hence $MEI_{MayJun}$ does not appear to be a long-lead predictor of snow accumulation, and so the differences in skill scores may be due to spurious correlations. Therefore we recommend both this model and the $Q_{March}$ and $MEI_{FebMar}$ model be compared and assessed for future events.*"

and have added the following finding to the conclusions:

"*For Tarbela the $Q_{March}$ and $MEI_{MayJun}$ model gave the best skill, however because we could not determine the dynamical mechanism(s) by which the relatively long lag between $MEI_{MayJun}$ influences snowpack accumulation and flow, we cannot rule out the possibility that the skill is due to spurious correlation. Therefore we recommend both this model and the $Q_{March}$ and $MEI_{FebMar}$ model be compared and assessed for future events and, more generally, that BMA be trialled in future research to combine the skill of multiple BJP models as, for example, undertaken in Australia in (Pokhrel et al., 2013)*"

(13) P8, L14-15. I thought that in some cases March flow was the highest skill predictor and adding any predictor didn't increase the skill so why are you using both March flow and climate predictors here?

**Response:** For Indus at Tarbela (Table 2), inclusion of a climate predictor in addition to the March flow predictor increased skill in all cases. For Jhelum at Mangla, there were mixed results as adding the climate predictor slightly decreased the skill for the full 1975-2015 period but increased the skill for the 2001-2015 period. As the 2001-2015 period is the comparison period for SRM, we think it is acceptable to use these predictor combinations.

(14) Figure 5 and 6. These figures are used to compare the skill of BJP vs SRM based streamflow forecasts. I think it would be better to combine them both into one figure. Maybe just show SRM forecasts with a different color.

**Response:** We are concerned that combining multiple forecast evaluations onto single figures would look cluttered in some circumstances, such as Panel (d), hence out preference is to maintain these figures as they are.

(15) Conclusion: The last two bullet points are not really findings. I suggest to separately discuss them after listing the findings. Also please mention here the current state-of-the-practice for generating streamflow forecasts in the region and the value the methods explored in this study will add.

**Response:** We agree and now present these last two bullet points as a separate paragraph after the list of findings. We have added a description of current methods, including discussion of the value added by our new approach. This modified text is as follows:

"*In future research, BJP forecast models could readily be developed and assessed for other tributaries, e.g. the Chenab and Kabul, subject to availability of flow data. This would allow an overall assessment of UIB flow forecasting for the major contributing basins. The present method used by the Indus River System Authority (IRSA) to forecast UIB Kharif streamflow is based on historical analogues. The IRSA use their database of the previous 60 years of flow to select years where the historic March flows are within 5% of the current March flow and use the corresponding historical Kharif flows (within 5%) for their forecast scenario. The selection of the historical scenario is also informed by forecasts from the Pakistan Meteorological Department, forecasts provided by WAPDA (e.g. SRM forecasts), and present snow conditions in the catchment. The forecasts are continuously revised as the season progresses.*

*Sufficiently skilful BJP forecasts could also inform scenario selection, providing for the first time a probabilistic approach to forecasts in contrast to a single forecast as currently used. However probabilistic forecasts (such as a the BJP) can be misinterpreted if they are unfamiliar to the water management professionals using them to inform decisions (Pagano et al., 2002;Ramos et al., 2013;Rayner et al., 2005;Whateley et al., 2015). Hence the successful transfer of BJP forecast tools to operational use within Pakistan would require guidance for building BJP models and generating forecasts, test cases with example results, face to face training, and on-going support.*"

**Response to interactive comment by Anonymous Referee #2**

The authors have assessed three different methods - Bayesian Joint Probability (BJP), the Snowmelt Runoff Model (SRM) and a hybrid approach (SRM - Ensemble Streamflow Prediction inflow means as additional predictor in BJP approach) for forecasting seasonal streamflow to the two largest dams in the Upper Indus Basin, Pakistan. The authors concluded that BJP approach is simple and it worked well to provide probabilistic seasonal streamflow forecasts. The topic is relevant for publication in HESS. Overall, the paper is well written. I recommend a moderate revision to the manuscript and the following concerns need to be addressed:

**Response:** Thank you for your encouraging and helpful review.

Major Concerns:

1. Under the BJP approach, was the conditional multivariate normal distributions fit over the entire season or on monthly basis? How many samples were generated through Monte Carlo Simulations under the BJP approach? Provide details.

**Response:** We have modified the text to include clarifying information, as follows:

*"The cross-validated BJP forecast performance was assessed for 1975-2015 (41 seasons), with the BJP models calibrated on a seasonal basis (i.e. 40 data points) using 1000 MCMC samples for each of the leave-one-out calibrations."*

2. Page 7, lines 24-27, the skill of using March flow and/or one climate predictor looks very similar to each other. The authors are recommended to use statistical significance test to compare if the skills are significantly different from each other.

**Response:** We have added uncertainty estimates to the skill scores Table 1 and Table 2, from bootstrapping, to aid interpretation of how different the forecast models are from one another. This leads to a discussion on model selection, with the addition of the following text to section 4.1 Skill scores:

*"Table 1 presents cross-validated BJP forecast skill scores using the trialled combinations of antecedent flow and climate predictors for the Kharif season for Jhelum at Mangla, together with bootstrapped $10^{th}$ to $90^{th}$ percentile ranges to assess model uncertainty. These ranges were obtained by resampling 1000 random sequences of years of the same length as the observed record, i.e. with replacement, and calculating skill scores for each sample."*

*"Given there is large uncertainty in skill scores we do not aim to select a 'best' model. However, as there are many models with positive skill (i.e. better than climatology) then using skilful models is plausible. Ideas on how to do this are discussed in section 5."*

and the following addition to section 5 Discussion:

*"More generally, the skill score uncertainty ranges presented in Table 1 and Table 2 highlight that no 'best' forecast model can be selected for either basin. Attempting to select a best model would ignore model uncertainty and thus not make best use of forecast skill across the range of models trialled. To address this, probabilistic forecasts from multiple BJP models can be combined using Bayesian Model Averaging to produce combined forecasts with higher skill than that obtainable from any individual model (Wang et al., 2012a). Thus trialling a BMA approach is recommended, although it is beyond the scope of this current work."*

3. Given that most of the streamflow at Indus River at Tarbela is snowmelt driven, use of a direct or indirect indicator of snow as one of the predictors, along with the projected summer air temperature can improve the forecasting skill. The authors are encouraged to consider global precipitation (for winter) and air temperature forecasts as predictors, which can represent snow as one of the inputs to the model.

**Response:** We agree that GCM forecasts of precipitation and temperature could potentially be used as predictors, however in this work we are assuming that the water resources practitioners do not readily have access to GCM seasonal climate forecast data (including hindcasts, needed for model establishment). Hence out approach relies on information regarding temperature and precipitation being captured by our selected climate index predictors (statistically). This could be the subject of future research with dynamical models, so we will mention this in our revised discussion.

4. It is not clear why MEI for May and Jun from previous year enhanced the skill score for Indus at Tarbela? Explain.

**Response:** We hypothesised that the MEI (May-June) predictor skill relates to autumn/winter snow accumulation, a lag of 4 months to snow accumulation from October onwards. It is thus not unreasonable that circulation systems bringing moisture into the region during autumn/winter are influenced by the forcing initiated by ENSO processes during the summer. We investigated this further and have added the following text to the discussion:

"*For Mangla, the predictor combination that gave the best Kharif season cross-validated skill scores included an ENSO-based predictor ($SSI_{March}$) immediately before the season (Table 1), which makes sense intuitively as it represents a climate driver of both the snow accumulation before and precipitation conditions during the Kharif season. In contrast, for Tarbela a much earlier ENSO-based predictor ($MEI_{MayJun}$, i.e. May-Jun the year before) provides higher skill scores than the equivalent predictor immediately before the season ($MEI_{FebMar}$) (Table 2). To try to understand the dynamical mechanism by which $MEI_{MayJun}$ is providing skill in forecasting $Q_{Kharif}$, we compared MEI correlations with GLDAS $SWE_{March}$, $Q_{March}$ and $Q_{Kharif}$. Results were inconclusive, and perhaps impeded by the short record lengths given SWE is only available from 2000, as while $MEI_{MayJun}$ has a higher correlation with $Q_{Kharif}$ than $MEI_{FebMar}$ (0.76 versus 0.63, respectively) it has a slightly lower correlation with $SWE_{March}$ (0.48 versus 0.52, respectively). Hence $MEI_{MayJun}$ does not appear to be a long-lead predictor of snow accumulation, and so the differences in skill scores may be due to spurious correlations. Therefore we recommend both this model and the $Q_{March}$ and $MEI_{FebMar}$ model be compared and assessed for future events.*"

and have added the following finding to the conclusions:

"*For Tarbela the $Q_{March}$ and $MEI_{MayJun}$ model gave the best skill, however because we could not determine the dynamical mechanism(s) by which the relatively long lag between $MEI_{MayJun}$ influences snowpack accumulation and flow, we cannot rule out the possibility that the skill is due to spurious correlation. Therefore we recommend both this model and the $Q_{March}$ and $MEI_{FebMar}$ model be compared and assessed for future events and, more generally, that BMA be trialled in future research to combine the skill of multiple BJP models as, for example, undertaken in Australia in (Pokhrel et al., 2013).*"

5. Page 7, line 1, how good or better the skill enhancement is if SSCRSP (or SSRMSE) changes from 21 to 24.3 (within moderate skill range in Table 1)? Does it reduce uncertainty? Clarify.

**Response:** As described in the verification methods section, improvements in CRPS reflect improvement in accuracy and/or sharpness and improvements in RMSE reflect improvements in accuracy of the median only. So an inference can be made through comparative analysis of the various skill metrics. We have added uncertainty estimates to the skill scores Table 1 and Table 2, from bootstrapping, to aid interpretation of how different the forecast models are from one another. This leads to a discussion on model selection, with the addition of the following text to section 4.1 Skill scores:

*"Table 1 presents cross-validated BJP forecast skill scores using the trialled combinations of antecedent flow and climate predictors for the Kharif season for Jhelum at Mangla, together with bootstrapped 10th to 90th percentile ranges to assess model uncertainty. These ranges were obtained by resampling 1000 random sequences of years of the same length as the observed record, i.e. with replacement, and calculating skill scores for each sample."*

*"Given there is large uncertainty in skill scores we do not aim to select a 'best' model. However, as there are many models with positive skill (i.e. better than climatology) then using skilful models is plausible. Ideas on how to do this are discussed in section 5."*

and the following addition to section 5 Discussion:

*"More generally, the skill score uncertainty ranges presented in Table 1 and Table 2 highlight that no 'best' forecast model can be selected for either basin. Attempting to select a best model would ignore model uncertainty and thus not make best use of forecast skill across the range of models trialled. To address this, probabilistic forecasts from multiple BJP models can be combined using Bayesian Model Averaging to produce combined forecasts with higher skill than that obtainable from any individual model (Wang et al., 2012a). Thus trialling a BMA approach is recommended, although it is beyond the scope of this current work."*

6. In Table 3, it will be good to know the correlations that are statistically significant (e.g. at 95% confidence interval) based on the sample size.

**Response:** We have added statistical significance in the table.

7. Page 10, lines 2-7, the hypotheses listed are not clear. As mentioned by the authors earlier, it is already known the snowmelt plays an important role for Indus River at Tarbela. So it not a hypothesis. Also, the results indicated that adding NAO, when used as a predictor, did not improve forecasting skill.

**Response:** We have re-worded to avoid the confusion caused by the term "hypothesised". Table 2 shows that the NAO predictor did add some skill, however not as much as the selected ENSO based predictor. The text now reads as follows:

*"These higher correlations in the late Kharif (relative to the early Kharif) for Indus at Tarbela would relate to the correspondingly higher relative skill scores shown in Figure 3 for late Kharif, corresponding to late-season glacier melt processes that are a significant component of the inflow to Tarbela but not Mangla (Mukhopadhyay and Khan, 2015).*

*It is also interesting to reflect on the relative performance of the NAO climate predictor, which does not provide any skill for inflow to Mangla (Table 1) but offers comparable skill to several of the ENSO indices trialled for Tarbela (Table 2). This indicates NAO may have some skill with regards to late season glacier melt. Overall, these results concur with investigations showing a stronger relationship between ENSO and precipitation and weaker relationship between NAO and precipitation in recent*

*decades (Yadav et al., 2009a;Yadav et al., 2009b) resulting in the prevalence of ENSO as the better predictor of winter snowpack magnitude.*"

Minor Concerns:

8. Did the models use monthly (or daily) data for the model fitting? If so, it needs to be clearly stated.

**Response:** The BJP is calibrated to seasonal data (i.e. 41 data points 1975 to 2015). This has been clarified in the text as follows:

"*The cross-validated BJP forecast performance was assessed for 1975-2015 (41 seasons), with the BJP models calibrated on a seasonal basis (i.e. 40 data points) using 1000 MCMC samples for each of the leave-one-out calibrations.*"

RC2:9. Page 6, lines 27 – 30, RMSEP needs to be used instead of RMSE. Also RMSEP needs to be defined in the text.

**Response:** We considered RMSEP but concluded similar relative results are obtained from RMSE, based on extensive experience across many previous studies (co-authors Wang, Schepen and Robertson). Hence we have continued with RMSE.

RC2:10. In figures 3a, 4a, 5a and 6a, what are the bounding lines (is it 95% Confidence Interval)?

**Response:** As stated in the figure caption, these are Kolmogorov 5% significance bands. We have clarified this in our revised text referring to these figures.

**Editor Decision: Reconsider after major revisions (further review by editor and referees)**
(03 Mar 2018)
by Andy Wood

Comments to the Author:
The authors' responses to reviewer concerns are appreciated, and the author should proceed in making the proposed corrections, with a few exceptions. In a number of key areas noted below, the authors can do more to strengthen the paper – primarily through (1) following reviewer suggestions further toward assessing additional, likely predictors from publically available data sources; and (2) further investigating and justifying the use of such a long-lag climate index (MEI). While it is well known that such indices provide predictability, such demonstrations have typically been at shorter lags (ie, values immediately prior to the prediction period). The authors must do more to defend the idea (through analysis) that predictability pathways exist, and that shorter lag indices (eg, the MEI itself, but in January of the same year as the prediction) cannot provide better skill. The current result is not well supported by existing literature on index-based hydroclimate prediction. Further notes on these points are given below.

**Response:** Thank you for providing recommendations to help strengthen the paper. We will undertake the proposed corrections and address the identified exceptions.

Regarding the comment "*The authors must do more to defend the idea (through analysis) that predictability pathways exist, and that shorter lag indices (eg, the MEI itself, but in January of the same year as the prediction) cannot provide better skill.*"
In Table 2 of the submitted manuscript, skills scores using the MEI for short-lag February-March (i.e. immediately before the forecast season) and long-lag May-June are both shown, both as individual predictors and in combination with $Q_{March}$. The skills scores show that the short-lag $MEI_{FebMar}$ provides considerably less skill than the selected long-lag $MEI_{MayJun}$. We acknowledge this is not sufficiently discussed and so have undertaken further analysis, as recommended, and have added the following to the discussion:

> "*For Mangla, the predictor combination that gave the best Kharif season cross-validated skill scores included an ENSO-based predictor ($SSI_{March}$) immediately before the season (Table 1), which makes sense intuitively as it represents a climate driver of both the snow accumulation before and precipitation conditions during the Kharif season. In contrast, for Tarbela a much earlier ENSO-based predictor ($MEI_{MayJun}$, i.e. May-Jun the year before) provides higher skill scores than the equivalent predictor immediately before the season ($MEI_{FebMar}$) (Table 2). To try to understand the dynamical mechanism by which $MEI_{MayJun}$ is providing skill in forecasting $Q_{Kharif}$, we compared MEI correlations with GLDAS $SWE_{March}$, $Q_{March}$ and $Q_{Kharif}$. Results were inconclusive, and perhaps impeded by the short record lengths given SWE is only available from 2000, as while $MEI_{MayJun}$ has a higher correlation with $Q_{Kharif}$ than $MEI_{FebMar}$ (0.76 versus 0.63, respectively) it has a slightly lower correlation with $SWE_{March}$ (0.48 versus 0.52, respectively). Hence $MEI_{MayJun}$ does not appear to be a long-lead predictor of snow accumulation, and so the differences in skill scores may be due to spurious correlations. Therefore we recommend both this model and the $Q_{March}$ and $MEI_{FebMar}$ model be compared and assessed for future events.*"

Response to Rev1-Q2/Rev2-Q3 – It would be worth assessing the skill of the NMME average at least (which is available operationally) over the region to see whether it can be included in a statistical framework as a predictor – even if it is not used in hydrological modeling. Even though the assessment recognizes limitations in the ability of practitioners to use data such as NMME, it should not be hard for the authors to extract the NMME data and evaluate whether it would be worth developing as an additional predictor.

**Response:** While we agree that assessing readily available dynamical forecasts of precipitation and temperature over the region could aid in identifying additional predictors, a recently published assessment of forecast skill over Pakistan and Afghanistan for NMME May 1 hindcasts (for May to

November) concluded that the MMEM, that generally exceeded the skill of any individual model, provides little benefit over climatology (Cash et al. 2017) [1].

Given this assessment, we would not expect NMME forecast precipitation or temperature for our study region to add skill as additional predictors to the BJP. We hope, however, that the usage of climate model forecasts can be investigated in the future in a more rigorous way and thus have added to the discussion the following:

 *"Future research could investigate whether dynamical seasonal forecasts of temperature have skill of relevance to forecasting glacier melt, however as noted above such skill has not been determined to date (e.g. Cash et al., 2017), and is beyond the scope of this assessment given our focus of developing practical and easily implementable forecast tools using readily available inputs."*

Response to Rev1-Q3/Rev2-Q3 -- Similarly, it would be worth following the reviewer's suggestion to see whether modeled SWE anomalies from GLDAS (also publicly available) would be a similarly useful predictor. Even if they have the same correlation with streamflow, if it is uncorrelated with antecedent flow, it may add skill. The reasons for not using MODIS are fair.

**Response:** We have investigated GLDAS SWE over our two study basins. Firstly there is the issue of record length, as GLDAS-2.0 only covers to 2010 (i.e. it isn't up to date and thus unsuitable for real-time operational forecasting) and GLDAS-2.1 only commences in 2000, i.e. a shorter period than the predictors assessed in the manuscript. Such a short period was also our justification for not using MODIS snow cover as a predictor, which similarly commenced in 2000.

Despite this short record length, we have assessed 2000-2015 annual correlation of v2.1 $SWE_{March}$ with Kharif (April-September) flow and $Q_{March}$ for our two study basins.
- For Jhelum at Mangla, $SWE_{March}$ and $Q_{March}$ correlation with Kharif flow are 0.50 and 0.73, respectively. Also, there is a 0.87 correlation between $SWE_{March}$ and $Q_{March}$, suggesting $SWE_{March}$ doesn't provide additional information and hence skill above that provided by $Q_{March}$.
- For Indus at Tarbella, $SWE_{March}$ and $Q_{March}$ correlation with Kharif flow are comparable at 0.56 and 0.55, respectively. The correlation between $SWE_{March}$ and $Q_{March}$ is 0.77, suggesting they are not independent predictors.

Therefore, the limitation of short record length, lack of higher correlation with flow than that of the $Q_{March}$ predictor, and relatively high cross-correlation with $Q_{March}$, leads us to conclude that GLDAS SWE would not provide additional skill as a predictor for the BJP. We've added text to reflect this to section 4.1 Skill scores, as follows:

*"MODIS (Hall et al., 2010) snow-cover area and GLDAS-2.1 (Rodell et al., 2004) snow-water equivalent, additional measures of antecedent conditions, were also assessed as potential predictors. A significant limitation is the shorter record lengths for MODIS and GLDAS, as available data for both start in 2000. This is of particular concern for the BJP's leave-one-out cross-validation, as using short records to identify dynamical mechanisms is susceptible to spurious skill. Correlation analysis, cognisant of the short 2000-15 period, found that these snow products have a similar or lower correlation with Kharif flow (QKharif) compared to March flow (QMarch), and are relatively highly correlated with QMarch. Thus the limitation of short record length, lack of higher correlation with flow than that of the QMarch predictor, and relatively high cross-correlation with QMarch, leads us to conclude that they would not be expected to provide additional skill as a predictor for the BJP."*

Response to Rev1-Q12/Rev2-Q4 – The significance of MEI as a predictor at a lag of nearly a year to the predictand still needs further justification. The Marriotti text does not adequately characterize the processes and potential predictability at such lags explicitly. Although one possible route toward this predictability could be, as suggested, the MEI prediction of fall winter climate variability, hence winter snow accumulation, presumably this could be more directly captured

[1] Cash, B. A., Manganello, J. V., and Kinter, J. L.: Evaluation of NMME temperature and precipitation bias and forecast skill for South Asia, Clim Dyn, 10.1007/s00382-017-3841-4, 2017.

using the actual variables (ie, winter snow peak, accumulated winter precip) than a long-lag teleconnection. Even the use of MEI at a shorter lag from the predictand would intuitively carry more information. This result suggests a spurious set of correlations without greater analysis of the underlying physical mechanisms. Could the paper establish whether the MEI is skillfully related to the proposed intermediate dynamics though additional analyses? Would using predictors such as winter precipitation anomalies provide similar predictability? Further analysis would result in a broader and more insightful study to benefit potential implementation.

**Response:**
Please see our first response above, addressing these points and outlining the additional analysis we have undertaken and the resulting discussion we've added to the paper.

[revised manuscript text omitted]